# Model-Informed Flows for Bayesian Inference

**Joohwan Ko,  Justin Domke**
Manning College of Information and Computer Sciences
University of Massachusetts Amherst
{joohwanko,domke}@cs.umass.edu

## Abstract

Variational inference often struggles with the posterior geometry exhibited by complex hierarchical Bayesian models. Recent advances in flow-based variational families and Variationally Inferred Parameters (VIP) each address aspects of this challenge, but their formal relationship is unexplored. Here, we prove that the combination of VIP and a full-rank Gaussian can be represented exactly as a forward autoregressive flow augmented with a translation term and input from the model's prior. Guided by this theoretical insight, we introduce the Model-Informed Flow (MIF) architecture, which adds the necessary translation mechanism, prior information, and hierarchical ordering. Empirically, MIF delivers tighter posterior approximations and matches or exceeds state-of-the-art performance across a suite of hierarchical and non-hierarchical benchmarks.

## 1  Introduction

Inference remains challenging for many complex Bayesian models. Variational Inference (VI) casts posterior approximation as minimization of the Kullback-Leibler divergence between a tractable approximating distribution and the true posterior [9, 25, 23]. Recent advances have focused on increasing the expressiveness of variational families. Flow-based models, which apply a sequence of invertible transformations to a simple base distribution, have shown particular promise [34, 36, 28].

Variationally Inferred Parameters (VIP) [19] address a key challenge in VI: The posteriors of hierarchical Bayesian models often exhibit pronounced curvature or "funnel-like" geometry [7, 24] that standard families like Gaussians cannot capture. VIP builds on insights from non-centered parameterization (NCP) in Markov chain Monte Carlo (MCMC) [35], adaptively learning the optimal degree of non-centering for each latent variable during the VI optimization process.

Our preliminary investigations reveal a striking observation: When VIP is combined with full-rank Gaussian variational families, it achieves performance comparable to state-of-the-art methods from recent evaluation [10] (See Tables 1 and  3). Since VIP and flow-based models both excel at navigating posterior geometry, might they be manifestations of a common principle? In particular, can a flow-based distribution encapsulate the benefits of VIP?

Our core theoretical result, formalized in Theorem 4, shows that every full-rank VIP transformation can be represented exactly as a forward autoregressive flow (FAF), provided that the flow (i) respects the topological order of the latent variables, (ii) augments the usual affine mapping with a new "translation" term, and (iii) receives the model's prior mean and scale functions as additional inputs.

Guided by these principles, we introduce the Model-Informed Flow (MIF), a forward autoregressive architecture that includes the necessary translation term and incorporates the distributional form of the model's prior. Across a range of benchmark models, MIF compares well to state-of-the-art VI methods. We also perform targeted ablations to verify the predicted implications and practical consequences of our theory.

39th Conference on Neural Information Processing Systems (NeurIPS 2025).

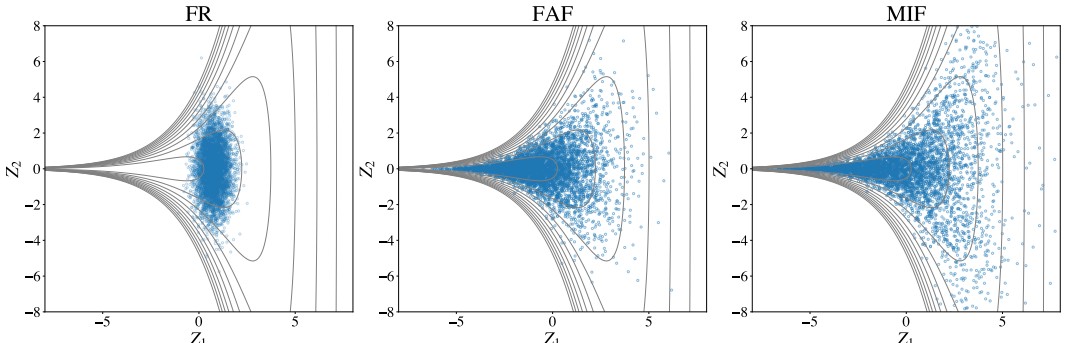

Figure 1: "Funnel"-type distributions commonly arise in hierarchical model. This figure shows the funnel distribution (gray contours) approximated with 5000 samples from three families (blue points): A Full-Rank Gaussian (FR) is very poor (KL-divergence of 1.86 nats). A standard Forward Autoregressive Flow (FAF) is much better (0.38 nats) but still imperfect. Our proposed model-informed flow (MIF) achieves a KL-divergence of effectively 0.

## 2 Preliminaries

### 2.1 Variational Inference

In Bayesian inference, given a model $p(z, x)$, the goal is to approximate the posterior distribution $p(z|x)$ given observed data $x$. Direct computation of this posterior is typically intractable, so variational inference (VI) introduces a tractable family $q_w(z)$ and minimises the Kullback–Leibler (KL) divergence from $q_w(z)$ to the true posterior. The marginal likelihood then decomposes as

$$\log p(x) = \underbrace{\mathbb{E}_{q_w(z)}\left[\log \frac{p(z,x)}{q_w(z)}\right]}_{\text{ELBO}\left[q_w(z) \,\|\, p(z,x)\right]} + \underbrace{\text{KL}\left[q_w(z) \,\|\, p(z|x)\right]}_{\text{divergence}}. \tag{2.1}$$

Since $\log p(x)$ is constant with respect to the variational parameters $w$, maximising the ELBO in (2.1) is equivalent to minimising the KL divergence between $q_w$ and the posterior.

### 2.2 Hierarchical Bayesian Models

Hierarchical Bayesian models are common in domains such as ecology [12, 43, 38] and epidemiology [16, 37, 29]. Following Gorinova et al. [19], we consider a hierarchical model in which each latent variable $z_i$ is conditionally Gaussian with a mean and variance that are arbitrary function of parent variables, while the observed data $x_j$ is generated from an arbitrary likelihood $p(x_j|\pi(x_j))$. In particular, the likelihood for $x_j$ may depend on one or more latent variables through its parent set $\pi(x_j)$. Formally, write

$$z_i \sim \mathcal{N}\Big(f_i\big(\pi(z_i)\big), g_i\big(\pi(z_i)\big)\Big), \quad x_j \sim p\big(x_j|\pi(x_j)\big), \tag{2.2}$$

where $\pi(\cdot)$ denotes the set of parent variables. Throughout, $\pi(\cdot)$ may include both observed and latent variables; that is, a variable can be conditioned on any subset of the remaining variables permitted by the model graph. In practice, the functions $f_i$ and $g_i$ (which return the mean and standard deviation of $z_i$, respectively) often arise directly from a probabilistic programming system, encoding how the mean and variance of each latent variable depend on its parent variables.

The joint density corresponding to Eq 2.2 is

$$p(z, x) = \prod_{i=1} \mathcal{N}\Big(z_i|f_i\big(\pi(z_i)\big), g_i\big(\pi(z_i)\big)\Big) \times \prod_{j=1} p\big(x_j|\pi(x_j)\big). \tag{2.3}$$

**Example.** As a concrete instance of Equation (2.2), consider the model

$$z_1 \sim \mathcal{N}(0, 1), \qquad\qquad z_2 \sim \mathcal{N}\big(\alpha z_1, \exp(\beta z_1)\big),$$
$$x_1 \mid z_1, z_2 \sim \text{Poisson}\big(\exp\big(z_1 + \tfrac{1}{2}z_2\big)\big), \qquad x_2 \mid z_2, x_1 \sim \text{Binomial}\big(n = x_1, p = \sigma(z_2)\big),$$

where $\sigma(u) = \frac{1}{1+\exp(-u)}$. To cast this as an instance of Equation (2.2), define the parent sets

$$\pi(z_1) = \emptyset, \quad \pi(z_2) = \{z_1\}, \quad \pi(x_1) = \{z_1, z_2\}, \quad \pi(x_2) = \{z_2, x_1\},$$

and the functions

$$f_1(\emptyset) = 0, \quad g_1(\emptyset) = 1, \qquad f_2(z_1) = \alpha z_1, \quad g_2(z_1) = \exp(\beta z_1).$$

In Equation (2.2), we make no assumptions about the distributional form of the observed variables $x_i$, so $p(x_1|\pi(x_1)) = \text{Poisson}\big(\exp(z_1 + \frac{1}{2}z_2)\big)$ and $p(x_2|\pi(x_2)) = \text{Binomial}(n = x_1, p = \sigma(z_2))$ can be arbitrary.

## 2.3 Non-Centered Parameterization

Hierarchical models often exhibit difficult curvature and "funnel-like" geometries. These are a challenge for both sampling and variational inference approaches [7, 24, 22]. A common strategy in MCMC to alleviate these issues is the *non-centered parameterization* (NCP) [35]. To illustrate, consider two latent variables $z_1$ and $z_2$ having prior of

$$z_1 \sim \mathcal{N}(\mu, \sigma), \quad z_2 \sim \mathcal{N}\big(0, \exp(z_1/2)\big). \tag{2.4}$$

In the standard (centered) parameterization, the goal is to approximate $p(z|x)$. However, the nonlinear dependence of $z_2$ on $z_1$ can yield a sharply funnel-shaped joint posterior. As illustrated in Figure 1 (left panel), a full-rank Gaussian struggles to capture this funnel structure.

The non-centered approach introduces auxiliary variables $\epsilon = (\epsilon_1, \epsilon_2) \sim \mathcal{N}(0, I)$ and defines a transformation $z = T_{\text{NCP}}(\epsilon)$ that recovers $(z_1, z_2)$ via

$$z_1 = \mu + \sigma \epsilon_1, \quad z_2 = \exp\left(\frac{z_1}{2}\right)\epsilon_2. \tag{2.5}$$

Instead of $p(z, x)$, inference is done on $p(\epsilon, x)$ where $p(\epsilon) = \mathcal{N}(0, I)$ and $p(x|\epsilon) = p(x|z = T_{\text{NCP}}(\epsilon))$. The intuition is that if $x$ is relatively uninformative, then $p(\epsilon|x)$ will be close to a standard Gaussian, even when $p(z|x)$ might have a funnel-type geometry.

## 2.4 Variationally Inferred Parameters

While NCP helps alleviate pathologies in hierarchical models, it is not ideal for every latent variable. Non-centering can prevent funnel-shaped geometries in cases with limited data, but when data is highly informative, a centered formulation may be better. *Variationally Inferred Parameters* (VIP) addresses this trade-off by learning a partial non-centered parameterization for each variable [19].

VIP reparameterizes $z \in \mathbb{R}^D$ by introducing an auxiliary variable $\tilde{z} = (\tilde{z}_1, \ldots, \tilde{z}_D)$, and by defining a parameter $\lambda_i \in [0, 1]$ per dimension. Rather than drawing $z_i$ directly from its prior, VIP samples

$$\tilde{z}_i \sim \mathcal{N}\left(\lambda_i f_i\left(\pi(z_i)\right), g_i\left(\pi(z_i)\right)^{\lambda_i}\right), \tag{2.6}$$

then recovers $z_i$ via VIP transformation, $T_{\text{VIP}}$, as in Definition 1. Note that $\lambda_i$ interpolates between the centered ($\lambda_i = 1$) and fully non-centered ($\lambda_i = 0$) extremes.

**Definition 1** (VIP Transformation). For a fixed ordering of latent variables $z = (z_1, \ldots, z_D)$, and auxiliary variables $\tilde{z} = (\tilde{z}_1, \ldots, \tilde{z}_D)$, let $\lambda_i \in [0, 1]$ be the partial non-centering parameter for each coordinate $i$, and $f_i(\pi(z_i))$ and $g_i(\pi(z_i))$ be continuous functions that return the mean and standard deviation of $z_i$ (respectively) based on the parent nodes $\pi(z_i)$. Then, the VIP transformation $z = T_{\text{VIP}}(\tilde{z})$ is defined coordinate-wise by

$$z_i = f_i(\pi(z_i)) + g_i(\pi(z_i))^{1-\lambda_i}\left(\tilde{z}_i - \lambda_i f_i(\pi(z_i))\right), \tag{2.7}$$

for $i = 1, \ldots, D$.

Substituting $T_{\text{VIP}}$ into the joint distribution (Equation (2.3)) yields a partially non-centered representation

$$p_{\text{VIP}}(\tilde{z}, x) = \prod_{i=1} \mathcal{N}\left(\tilde{z}_i | \lambda_i f_i\big(\pi(z_i)\big), g_i\big(\pi(z_i)\big)^{\lambda_i}\right) \times \prod_{j=1} p\big(x_j | \pi(x_j)\big). \tag{2.8}$$

VIP then learns a variational distribution $q_w(\tilde{z})$ by minimizing $\text{KL}\left(q_w(\tilde{z}) \| p_{\text{VIP}}(\tilde{z} \mid x)\right)$, optimizing both the variational parameters $w$ and the partiality parameters $\lambda$.

## 2.5 Equivalence of VIP in Model vs Variational Spaces

VIP applies $T_{\text{VIP}}$, in the *model space*. However, there is a straightforward analog in the *variational space*. Suppose $q_w(\tilde{z})$ is any distribution over the non-centered latent variables $\tilde{z}$. We define a new distribution $q_{w,\text{VIP}}(z)$ by letting

$$z = T_{\text{VIP}}(\tilde{z}, x) \quad \text{with} \quad \tilde{z} \sim q_w(\tilde{z}). \tag{2.9}$$

In other words, $q_{w,\text{VIP}}$ is the distribution induced on $z$ by applying the VIP transformation $T_{\text{VIP}}$ to samples from $q_w$. Concretely, its density follows from the change-of-variables formula

$$q_{w,\text{VIP}}(z) = q_w\left(T_{\text{VIP}}^{-1}(z, x)\right) \left|\det \nabla_{\tilde{z}} T_{\text{VIP}}(\tilde{z}, x)\right|^{-1}. \tag{2.10}$$

Since the KL divergence is invariant under such invertible transformations [11], we have that

$$\text{KL}\left(q_{w,\text{VIP}}(z) \| p(z|x)\right) = \text{KL}\left(q_w(\tilde{z}) \| p_{\text{VIP}}(\tilde{z}|x)\right). \tag{2.11}$$

Thus, optimizing $\min_{w,\lambda} \text{KL}\left(q_{w,\text{VIP}}(z) \| p(z|x)\right)$ with the posterior fixed is equivalent to reparameterizing the posterior as in 2.4. In this paper, we choose a Gaussian distribution for $q_w(\tilde{z})$ following the original framework [19]. In practice, we can choose any base family $q_w(\tilde{z})$ and still reap the benefits of partial non-centering. For more details please refer to Algorithm 2 in the Appendix B.

## 2.6 Forward Autoregressive Flows

Flows offer a powerful way to extend the expressiveness of variational families, thereby reducing the KL divergence between the variational approximation and the true posterior [36, 28, 2, 1]. In flows, a simple base distribution is gradually transformed into a more complex one through a sequence of invertible mappings. One such transformation is the forward autoregressive flow.

**Definition 2.** A *forward autoregressive flow* (FAF) is an invertible transformation $z = T_{\text{FAF}}(\epsilon)$, which maps a base random variable $\epsilon = (\epsilon_1, \ldots, \epsilon_D) \sim p_\epsilon$ to the target variable $z = (z_1, \ldots, z_D)$ via an autoregressive transformation

$$z_i = m_i(z_{1:i-1}) + s_i(z_{1:i-1})\,\epsilon_i, \quad i = 1, \ldots, D, \tag{2.12}$$

where $p_\epsilon$ is a base distribution, $m_i$ is a shift function and $s_i$ is a scaling function. Furthermore, an *affine* forward autoregressive flow is one in which $m_i$ and $\log s_i$ are affine functions.

In practice, the scaling function is often parameterized in terms of $\log s_i$ and subsequently recovered using the exponential function, which ensures positivity; however, alternative nonlinear functions can also be used. Also, multiple layers of the transformation can be composed to build more flexible variational families.

## 3 Model-Informed Flows: Bridging VIP and Autoregressive Flows

We return to our central question: *Can a flow-based distribution encapsulate the benefits of VIP?* The VIP process, when applied in the variational space, involves two key stages: first, sampling an auxiliary variable $\tilde{z}$ from a base variational distribution $q_w(\tilde{z})$, and second, transforming $\tilde{z}$ into the latent variable $z$ using the VIP transformation, $T_{\text{VIP}}$.

The original VIP framework [19] primarily considers $q_w(\tilde{z})$ to be a mean-field Gaussian (i.e., with a diagonal covariance matrix). Our investigation extends this by considering $q_w(\tilde{z})$ as a full-rank Gaussian.

We first establish that a full-rank Gaussian is a special case of FAF. A full-rank Gaussian variational distribution $\tilde{z} \sim \mathcal{N}(\mu, LL^\top)$ is generated by an affine transformation $T_A$, applied to a standard normal variable $\epsilon \sim \mathcal{N}(0, I)$

$$\tilde{z} = T_A(\epsilon) = \mu + L\,\epsilon, \tag{3.1}$$

where $\mu \in \mathbb{R}^D$ is the mean vector and $L \in \mathbb{R}^{D \times D}$ is a lower-triangular Cholesky factor of the covariance matrix. This affine transformation, $T_A$, can be replicated by an affine FAF by defining the shift and scale functions of the FAF (recall Definition 2) as

$$m_i(\tilde{z}_{1:i-1}) = \mu_i + L_{i,1:i-1} L_{1:i-1,1:i-1}^{-1}(\tilde{z}_{1:i-1} - \mu_{1:i-1}), \quad \log s_i(\tilde{z}_{1:i-1}) = \log L_{ii}, \tag{3.2}$$

(see Lemma 7 in Appendix C). Thus, a full-rank Gaussian distribution can be viewed as a specific instance of an affine FAF.

While a standard affine FAF can represent the initial transformation $T_A$ (generating $\tilde{z}$ from $\epsilon$), it cannot, by itself, fully represent the composite transformation $T_{\text{VIP}} \circ T_A$ that yields $z$. To see why, observe that the composite transformation reduces to

$$z_i = f_i(\pi(z_i)) + g_i(\pi(z_i))^{1-\lambda_i} \left( \mu_i + \sum_{j=1}^{i-1} L_{ij}\epsilon_j + L_{ii}\epsilon_i - \lambda_i f_i(\pi(z_i)) \right). \tag{3.3}$$

If we attempt to cast this composite transformation into the standard affine FAF form from Equation (2.12) the terms would need to be

$$m_i(z_{1:i-1}) \overset{?}{=} f_i(\pi(z_i)) + g_i(\pi(z_i))^{1-\lambda_i} \left( \mu_i + \sum_{j=1}^{i-1} L_{ij}\epsilon_j - \lambda_i f_i(\pi(z_i)) \right), \tag{3.4}$$

$$\log s_i(z_{1:i-1}) \overset{?}{=} \log g_i(\pi(z_i))^{1-\lambda_i} + \log L_{ii}. \tag{3.5}$$

In principle, sufficiently powerful networks in the FAF could represent these forms. However, they could not be captured in an affine FAF due to dependence on $\epsilon$ and prior functions. In addition, Equation (3.4) contains the product $g_i(\pi(z_i)) f_i(\pi(z_i))$, which induces quadratic dependence on $z$ that an affine mapping cannot represent. These nonlinearities may be difficult to capture even with an FAF that uses, e.g., multi-layer perceptrons. To remedy this, we introduce a generalized flow.

**Definition 3.** A *generalized forward autoregressive flow* (GFAF) is an invertible transformation $z = T_{\text{GFAF}}(\epsilon)$, which maps a base random variable $\epsilon = (\epsilon_1, \ldots, \epsilon_D) \sim p_\epsilon$ to the target variable $z = (z_1, \ldots, z_D)$ via an autoregressive transformation

$$z_i = m_i(z_{1:i-1}) + s_i(z_{1:i-1})\big(\epsilon_i - t_i(\epsilon_{1:i-1}, z_{1:i-1})\big), \quad i = 1, \ldots, D, \tag{3.6}$$

where $p_\epsilon$ is a base distribution, $m_i$ is a shift function, $s_i$ is a scaling function, and $t_i$ is a translation function. Furthermore, an *affine* generalized forward autoregressive flow is one in which $m_i$, $\log s_i$, and $t_i$ are affine functions.

Our main theoretical result, stated in Theorem 4, demonstrates how the composite transformation $T = T_{\text{VIP}} \circ T_A$ (a full-rank Gaussian transformation followed by the VIP transformation) can be exactly represented by such a generalized forward autoregressive flow.

**Theorem 4.** *Let $T = T_{\text{VIP}} \circ T_A$ where $T_{\text{VIP}}$ is the VIP transformation (Definition 1) and $T_A$ is the affine transformation from full-rank Gaussian. If $f_i$ and $\log g_i$ in the hierarchical Bayesian model (Equation (2.2)) are arbitrary continuous functions, and the parent nodes $\pi(z_i)$ are a subset of the preceding variables $z_{1:i-1}$ (respecting the causal dependency structure of the model), then $T$ can be represented as an affine generalized forward autoregressive flow with*

$$m_i(z_{1:i-1}) = f_i(\pi(z_i)), \tag{3.7}$$

$$\log s_i(z_{1:i-1}) = \log L_{ii} + (1 - \lambda_i) \log g_i(\pi(z_i)), \tag{3.8}$$

$$t_i(\epsilon_{1:i-1}, z_{1:i-1}) = \frac{1}{L_{ii}} \left( \lambda_i f_i(\pi(z_i)) - \mu_i - L_{i,1:i-1}\epsilon_{1:i-1} \right) \tag{3.9}$$

Theorem 4 therefore demonstrates that GFAF, by leveraging its novel translation term $t_i$ (which depends on past noise $\epsilon_{1:i-1}$) and by incorporating the model's prior functions, can replicate the full-rank VIP mechanism.

## 3.1 Model-Informed Flows

Building on the previous ideas, we introduce a simple generalization of FAFs designed to capture the VIP mechanism even if the conditioning networks are affine.

**Definition 5.** A *Model-Informed Flow* (MIF) is an invertible transformation $z = T_{\text{MIF}}(\epsilon)$, which maps a base random variable $\epsilon = (\epsilon_1, \ldots, \epsilon_D) \sim p_\epsilon$ to the target variable $z = (z_1, \ldots, z_D)$ via an autoregressive transformation

$$z_i = m_i(u_i) + s_i(u_i)(\epsilon_i - t_i(\epsilon_{1:i-1}, u_i)), \quad i = 1, \ldots, D, \tag{3.10}$$

$$u_i = [z_{1:i-1}, \ f_i(\pi(z_i)), \ g_i(\pi(z_i))] \tag{3.11}$$

---

**Algorithm 1** Model-Informed Flow (MIF)

---

**Input:** $\epsilon = (\epsilon_1, \cdots, \epsilon_D)$, prior functions $f_1, \cdots, f_D$ and $g_1, \cdots, g_D$ (or their logs for scale)

1: **for** $i = 1$ to $D$ **do**  ▷ Process in a pre-defined topological order of latent variables
2:   $u_i \leftarrow [z_{1:i-1}, f_i(\pi(z_i)), \log g_i(\pi(z_i))]$  ▷ Construct input incorporating prior information
3:   $m_i \leftarrow \text{NN}_m(u_i)$  ▷ Neural network for the shift function
4:   $\log s_i \leftarrow \text{NN}_s(u_i)$  ▷ Neural network for the log-scale function
5:   $t_i \leftarrow \text{NN}_t([u_i, \epsilon_{1:i-1}])$  ▷ Neural network for the translation term with $\epsilon_{1:i-1}$ inputs
6:   $z_i \leftarrow m_i + \exp(\log s_i) \cdot \left(\epsilon_i - t_i\right)$  ▷ Generalized autoregressive transformation
7: **end for**
8: **return** $z = (z_1, \ldots, z_D)$

---

where $p_\epsilon$ is a base distribution, $m_i$ is a shift function, $s_i$ is a scaling function, and $t_i$ is a translation term. Furthermore, an *Affine* Model-Informed Flow is one in which $m_i$, $\log s_i$, and $t_i$ are affine functions.

Algorithm 1 presents the pseudocode for implementing MIF, highlighting in blue the components that extend a standard forward autoregressive flow; omitting these reduces MIF to such a standard FAF. The key architectural differences are, first, MIF employs conditioning networks not only for the shift $m_i$ and log-scale $\log s_i$ but also for the translation term $t_i$, with the latter explicitly taking previous noise variables $\epsilon_{1:i-1}$ as inputs, as motivated by Theorem 4. Second, MIF is designed to incorporate the model's prior functions $f_i$ and $\log g_i$ as additional inputs to these conditioning networks, providing them with valuable, explicit structural guidance from the target model. Third, the generation process in MIF (the loop over $i$) must adhere to a pre-defined topological order of the latent variables consistent with the hierarchical Bayesian model.

Before moving on to experiments, we discuss two practical issues.

## 3.2 Incorporating Prior Information through $f_i$ and $\log g_i$

Our preliminary experiments revealed that omitting explicit prior function inputs $(f_i, \log g_i)$ from MIF often minimally impacted performance. This occurs when these priors are affine, as the target generalized forward autoregressive flow (GFAF) representing full-rank VIP (per Theorem 4) then simplifies to an *affine* GFAF. An MIF, even without these explicit prior inputs, can learn to reproduce this affine GFAF if its own conditioning functions for $m_i, \log s_i$, and $t_i$ possess at least affine capacity. This is shown in the following corollary.

**Corollary 6.** *Let $T = T_{\text{VIP}} \circ T_A$ where $T_{\text{VIP}}$ is the VIP transformation (see Definition 1) and $T_A$ is the affine transformation from full-rank Gaussian. If $f_i$ and $\log g_i$ in the hierarchical Bayesian model (see Equation (2.2)) are affine functions, then $T$ can be represented as an affine generalized forward autoregressive flow.*

This corollary shows that when the prior functions are affine, affine MIFs have enough capacity to learn them even if they are not provided, meaning there is no impact on the optimal KL-divergence. Still, in practice providing $f_i$ and $\log g_i$ is helpful in practice to accelerate convergence and reduce training complexity. We suspect the same pattern holds more generally—if a MIF is trained with high-capacity networks, it may be able to learn to represent the prior functions if not provided, at some extra training cost.

## 3.3 The Necessity of Conditioning on Previously Generated Latent Variables

Inverse Autoregressive Flows (IAFs) [28] represent an important class of normalizing flows, lauded for the computational efficiency of their sampling process. Unlike FAFs, conditioning networks of IAFs depend solely on the base noise variables $\epsilon_{1:i-1}$, allowing parallel computation of all $z_i$:

$$z_i = m_i(\epsilon_{1:i-1}) + s_i(\epsilon_{1:i-1})\epsilon_i, \quad i = 1, \ldots, D. \tag{3.12}$$

This structure makes it challenging to accurately represent the VIP mechanism. To capture VIP, an IAF's conditioning networks would need to replicate target forms that intrinsically depend on previously generated latent variables $z_{1:i-1}$ via the model's prior functions ($f_i(\pi(z_i))$ and $g_i(\pi(z_i))$).

Table 1: Negative ELBOs ($-$ELBO) for hierarchical benchmark models using mean-field Gaussian (MF), mean-field Gaussian with VIP (MF-VIP), full-rank Gaussian (FR), and full-rank Gaussian with VIP (FR-VIP). Lower values indicate tighter posterior approximations.

| Model | MF | MF-VIP | FR | FR-VIP |
|---|---|---|---|---|
| 8Schools | 34.80 | 31.89 | 33.85 | **31.86** |
| Credit | 548.65 | 533.90 | 535.88 | **525.02** |
| Funnel | 1.86 | 0.00 | 1.86 | **0.00** |
| Radon | 1267.99 | 1215.17 | 1220.73 | **1213.52** |
| Movielens | 870.93 | 850.49 | 856.11 | **844.51** |
| IRT | 823.15 | 822.58 | 817.38 | **816.64** |

Specifically, they would need to satisfy

$$m_i(\epsilon_{1:i-1}) \stackrel{?}{=} f_i(\pi(z_i)) + g_i(\pi(z_i))^{1-\lambda_i}\Big(\mu_i + \sum_{j=1}^{i-1} L_{ij}\epsilon_j - \lambda_i f_i(\pi(z_i))\Big), \tag{3.13}$$

$$\log s_i(\epsilon_{1:i-1}) \stackrel{?}{=} (1-\lambda_i)\log g_i(\pi(z_i)) + \log L_{ii}, \tag{3.14}$$

where $z_i$ (and thus $\pi(z_i)$) on the right-hand side are themselves complex functions of $\epsilon_{1:i-1}$. Asking $m_i(\epsilon_{1:i-1})$ and $s_i(\epsilon_{1:i-1})$ to represent this transformation is challenging as it essentially requires the network to reproduce the entire process that produced $z_{1:i-1}$ in previous steps.

Particularly, an IAF with *affine* conditioning networks for $m_i, \log s_i$ (and a term like $t_i$, if added) cannot capture the VIP transformation, since $z_{1:i-1}$ and $\epsilon_{1:i-1}$ have a complex non-linear relationship. This limitation underscores the advantage of forward autoregressive structures. We show in our experiments (e.g., results related to Figure 2) that IAFs require substantially greater model capacity to approach the effectiveness of MIF in representing VIP.

## 4 Related Work

Early flow-based generative models—including MADE [18], Real NVP [15], Glow [27], and Masked Autoregressive Flow [34]—established that autoregressive masking, coupling transforms, and invertible $1\times1$ convolutions enable tractable likelihoods with strong high-dimensional density estimation. Normalizing flows were subsequently adapted for variational inference, first as post-hoc enrichments of diagonal Gaussians [36] and later as architectures that scale in latent dimensionality, most notably Inverse Autoregressive Flow (IAF) [28]. These developments demonstrated that flexible, invertible parameterizations can substantially tighten the ELBO without sacrificing computational efficiency.

For hierarchical Bayesian models, *Variationally Inferred Parameters* (VIP) introduced gradient-based partial non-centering to alleviate the funnel geometry typical of deep hierarchies [19]. Building on VIP's idea of injecting model structure into the variational family, hybrids such as Automatic Structured Variational Inference (ASVI) [3], Cascading Flows [4], and Embedded-Model Flows (EMF) [41] couple model-aware parameterizations with flow flexibility. Our *Model-Informed Flow* continues this line by proving that full-rank VIP can be represented in a forward autoregressive flow augmented with translation and prior inputs, yielding a compact yet expressive variational family for hierarchical models.

## 5 Experiments

### 5.1 Experimental Setup

To validate our theoretical results, we evaluate on six different hierarchical Bayesian models: 8Schools, German Credit, Funnel, Radon, Movielens, and IRT. These models exhibit varying degrees of funnel-like posterior geometries, making them ideal testbeds for examining the benefits of VIP-inspired flow designs. For our final comprehensive benchmark experiments, we also include more models such as Seeds, Sonar, and Ionosphere.

Table 2: Negative ELBOs ($-$ELBO) for ablation study comparing affine Model-Informed Flow (MIF) against variations and baselines across hierarchical models. MIF includes z-conditioning, translation term, prior info, and correct order. Please refer to Appendix D for more details of variants of MIF. Lower values indicate tighter posterior approximations.

| Model | FR-VIP | MIF | MIF ($\epsilon$-cond) | MIF (w/o $t_i$) | MIF (w/o Prior) | MIF (w/o Order) | IAF |
|---|---|---|---|---|---|---|---|
| 8Schools | 31.86 | **31.74** | 32.04 | 31.86 | 31.83 | 33.83 | 32.22 |
| Credit | 525.02 | **520.72** | 523.92 | 525.87 | 522.75 | 534.63 | 523.85 |
| Funnel | **0.00** | 0.01 | 0.38 | 0.38 | 0.38 | 1.86 | 0.37 |
| Radon | 1213.52 | **1213.11** | 1215.11 | 1214.31 | 1213.45 | 1259.13 | 1215.23 |
| Movielens | 844.51 | **842.91** | 844.01 | 846.66 | 843.59 | 854.10 | 844.23 |
| IRT | 816.64 | **815.49** | 815.70 | 815.74 | 815.53 | 815.58 | 815.66 |

For all experiments, we utilize the Adam optimizer [26] and initialize all learnable parameters from a standard Gaussian distribution with a standard deviation of 0.1. For our Model-Informed Flow (MIF) evaluations, distinct configurations are used: the experiments in Section 5.3 primarily consider a single-layer *affine* MIF to clearly assess its structural components. In subsequent experiments, including the network expressiveness analysis and comprehensive benchmarks, MIF's conditioning networks are implemented as multi-layer perceptrons with ReLU activation functions to explore the impact of increased representational capacity via hidden units.

We perform 100,000 optimization iterations, using 256 Monte Carlo samples to approximate the ELBO during training. After training, the final ELBO is evaluated using 100,000 fresh samples for reliable estimation. For each model configuration, we report the best ELBO achieved after exploring six learning rates (logarithmically spaced from $10^{-1}$ to $10^{-6}$). Additional details regarding the benchmark models, specific variational family configurations, training time comparison, and MLP architectures are provided in Appendix D. Our implementation is available at `https://github.com/joohwanko/Model-Informed-Flow`

## 5.2 Preliminary Observations: Full-Rank VIP Outperforms Baselines

Prior work on VIP focused exclusively on mean-field Gaussian approximations [19]. To assess whether the advantages of VIP extend to richer variational families, we apply VIP to a full-rank Gaussian setting and compare four configurations—MF, MF-VIP, FR, and FR-VIP—across six canonical hierarchical models. As shown in Table 1, FR-VIP achieves the lowest negative ELBO in every case, outperforming both its non-VIP counterpart and mean-field VIP, and achieving performance comparable to more complex state-of-the-art methods (cf. Table 3). These results highlight the power of combining full-rank with VIP and motivate our design of VIP-inspired low architectures.

## 5.3 Impact of Design Components

In this section, we empirically validate the contribution of MIF's distinct design elements through a targeted ablation study, with results presented in Table 2. We compare the affine MIF against variants that individually alter or remove its key theoretically-motivated components: the $\epsilon$-dependent translation term ($t_i$), by evaluating MIF (w/o $t_i$); the use of explicit prior information, by assessing MIF (w/o Prior) which omits $f_i, \log g_i$ as inputs; and adherence to the hierarchical variable processing order, examined via MIF (w/o Order). Additionally, we test an IAF-like variant, MIF ($\epsilon$-cond), which conditions primarily on base noise $\epsilon_{1:i-1}$ instead of previously generated latents $z_{1:i-1}$, alongside relevant baselines like FR-VIP and a standard IAF. The consistent outcome is that deviating from the full MIF design by removing or altering these components generally degrades performance, underscoring their collective importance to its effectiveness.

## 5.4 Impact of Network Expressiveness

Recall from Section 3.3 that while IAFs face fundamental challenges in representing the VIP mechanism due to their conditioning solely on noise ($\epsilon$), it was posited that sufficient network expressiveness might partially overcome these limitations. We empirically investigate this by comparing our full

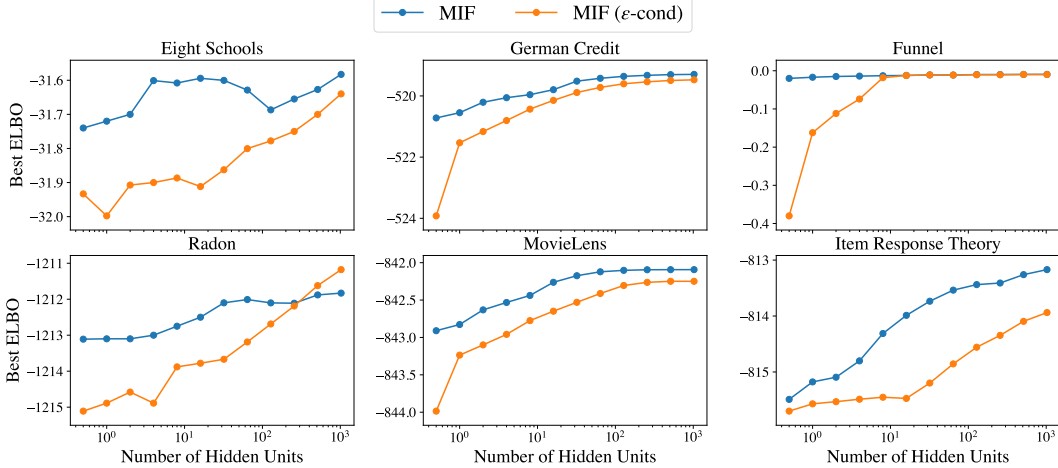

Figure 2: Best ELBO achieved by full MIF (blue) and the variant without latent-variable conditioning (orange) as a function of MLP hidden-unit count. Higher capacity allows the no-latent variant to close the gap, showing that sufficiently expressive networks can implicitly learn dependencies otherwise provided by explicit latent inputs.

MIF, which conditions on previously generated latents $z_{1:i-1}$, against MIF ($\epsilon$-cond), an IAF-like variant where the conditioning networks primarily use $\epsilon_{1:i-1}$ instead of $z_{1:i-1}$.

Figure 2 illustrates the performance of MIF versus MIF ($\epsilon$-cond) as the expressiveness of their conditioning networks (e.g., by increasing the number of hidden units) is varied. Across most benchmarked models, enhancing network capacity allows MIF ($\epsilon$-cond) to substantially improve its ability to capture complex dependencies. Consequently, the performance gap between this IAF-like variant and the standard MIF significantly narrows with increased expressiveness, demonstrating that sufficiently powerful IAF-like architectures can indeed achieve strong results, in many cases closely approaching or nearly matching the performance of MIF, which itself may maintain a slight lead or offer benefits in parameter efficiency.

### 5.5 Comprehensive Evaluation on Hierarchical and Non-hierarchical Benchmarks

To assess its generality and competitive strength, we benchmark MIF with 1024 hidden units per conditioning network, denoted MIF($h = 2^{10}$), against several leading variational inference methods. These baselines, primarily drawn from the recent benchmark by [10] and detailed with their acronyms in the caption of Table 3, provide a robust comparison. The evaluation spans the six previously discussed hierarchical models and three additional datasets (Seeds[†], Sonar[†], Ionosphere[†]) to test MIF's broad applicability. Results in the Table 3 show that MIF($h = 2^{10}$) achieves the best or equal-best performance on most of the benchmarks. These strong results, obtained using just a single layer of our expressive MIF transformation, demonstrate that its theory-driven design effectively delivers state-of-the-art performance with notable architectural simplicity.

## 6 Discussion and Limitation

This paper introduces the Model-Informed Flow (MIF), a new architecture for variational inference designed for complex hierarchical Bayesian models. MIF originates from a connection between VIP and FAFs. A significant finding is that even a single-layer affine MIF can achieve surprisingly competitive performance against other methods. A limitation of MIF is its reliance on the FAF structure. Although FAFs may not inherently demand more floating-point operations (FLOPs) per sample than Inverse Autoregressive Flows (IAFs), with variational inference, IAFs are better suited for modern parallel computing environments. The VIP mechanism appears to be essentially sequential in nature. Thus, our research suggests a consideration for flows and VI: to capture difficult posterior geometry may require either sequential processing as in MIF or high-capacity conditioning networks.

Table 3: Negative ELBOs ($-$ELBO) for various VI methods on hierarchical and non-hierarchical benchmarks. Compared methods include: Gaussian Mean-Field (MF) [8], Gaussian Mixture Model VI (GMMVI) [6], Sequential Monte Carlo (SMC) [14], Annealed Flow Transport (AFT) [5], Flow Annealed IS Bootstrap (FAB) [32], Continual Repeated AFT (CRAFT) [31], and Uncorrected Hamiltonian Annealing (UHA) [42]. Lower values indicate tighter posterior approximations; [†] denotes results taken from [10].

| Model | MF | GMMVI | SMC | AFT | FAB | CRAFT | UHA | MIF($h = 2^{10}$) |
|---|---|---|---|---|---|---|---|---|
| 8Schools | 34.80 | **31.70** | 31.80 | 31.81 | **31.79** | 31.90 | 33.09 | **31.78** |
| German Credit | 548.65 | 519.35 | 531.42 | 527.60 | 519.39 | 524.59 | 536.30 | **519.29** |
| Funnel | 1.86 | **0.01** | 0.29 | 0.30 | **0.01** | 0.02 | 0.38 | **0.00** |
| Radon | 1267.99 | 1213.67 | 1221.63 | 1236.42 | 1215.92 | 1229.77 | 1260.38 | **1211.83** |
| Movielens | 870.93 | 848.00 | 849.68 | 853.84 | 845.32 | 850.20 | 861.22 | **842.09** |
| IRT | 823.15 | 815.05 | 820.46 | 819.56 | 816.44 | 818.66 | 821.84 | **813.17** |
| Seeds[†] | 76.73 | 73.41 | 74.69 | 74.26 | 73.41 | 73.79 | N/A | **72.00** |
| Sonar[†] | 137.67 | 108.72 | 111.35 | 110.67 | **108.59** | 115.61 | N/A | 108.85 |
| Ionosphere[†] | 123.41 | 111.83 | 114.75 | 113.27 | **111.67** | 112.38 | N/A | 111.85 |

# 7 Acknowledgement

This material is based upon work supported in part by the National Science Foundation under Grant No. 2045900.

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

# Supplementary Materials

## A    Acronyms and Notations

This section provides a summary of acronyms and mathematical notations frequently used throughout the paper to aid the reader.

Table 4: List of Acronyms

| Acronym | Definition |
|---------|------------|
| CP | Centered Parameterization |
| ELBO | Evidence Lower Bound |
| FAF | Forward Autoregressive Flow |
| FR | Full-Rank (Gaussian) |
| GFAF | Generalized Forward Autoregressive Flow |
| IAF | Inverse Autoregressive Flow |
| KL | Kullback-Leibler (divergence) |
| MF | Mean-Field (Gaussian) |
| MIF | Model-Informed Flow |
| MLP | Multi-Layer Perceptron |
| NCP | Non-Centered Parameterization |
| VI | Variational Inference |
| VIP | Variationally Inferred Parameters |

Table 5: List of Notations

| Symbol | Definition |
|--------|------------|
| $x$ | Observed data |
| $z$ | Latent variables, $z = (z_1, \ldots, z_D)$ |
| $D$ | Dimensionality of the latent space |
| $\pi(z_i)$ | Set of parent variables for $z_i$ in a hierarchical model |
| $f_i(\cdot)$ | Prior mean function for $z_i$, i.e., $z_i \sim \mathcal{N}(f_i(\pi(z_i)), \cdot)$ |
| $g_i(\cdot)$ | Prior standard deviation function for $z_i$, i.e., $z_i \sim \mathcal{N}(\cdot, g_i(\pi(z_i)))$ |
| $\epsilon$ | Base noise variable, typically $\epsilon \sim \mathcal{N}(0, I)$ |
| $T_{\mathrm{NCP}}(\cdot)$ | Non-Centered Parameterization transformation |
| $\tilde{z}$ | Auxiliary base variable in VIP/NCP, $z = T(\tilde{z})$ |
| $\lambda_i$ | Partial non-centering parameter for $z_i$ in VIP, $\lambda_i \in [0, 1]$ |
| $T_{\mathrm{VIP}}(\cdot)$ | VIP transformation |
| $T_A(\cdot)$ | Affine transformation, e.g., $\mu + L\epsilon$ for a Gaussian |
| $m_i(\cdot)$ | Shift function for $z_i$ in an autoregressive flow |
| $s_i(\cdot)$ | Scale function for $z_i$ in an autoregressive flow |
| $t_i(\cdot)$ | Translation function for $\epsilon_i$ in GFAF/MIF |
| $u_i$ | Input to conditioning networks in MIF: $[z_{1:i-1}, f_i(\pi(z_i)), g_i(\pi(z_i))]$ |

## B    Algorithm

Algorithm 2 details the procedure for generating a sample $z$ from the variational distribution $q_{w,\mathrm{VIP}}(z)$ and then evaluating its corresponding log-density.

---

**Algorithm 2** Sampling from and Evaluating $q_{w,\text{VIP}}(z)$

---

**Input:** Parameters $w$ of the base distribution $q_w(\tilde{z})$; VIP parameters $\lambda_1, \ldots, \lambda_D$; model prior functions $f_i(\cdot)$, $g_i(\cdot)$, and parent dependencies $\pi(\cdot)$; dimensionality $D$; a topological ordering for $z_1, \ldots, z_D$.

1: Sample $\tilde{z} \sim q_w(\tilde{z})$
2: $\log J \leftarrow 0$
3: **for** $i = 1, \ldots, D$ **do**
4: $\quad z_i \leftarrow f_i\big(\pi(z_i)\big) + g_i\big(\pi(z_i)\big)^{1-\lambda_i}\big(\tilde{z}_i - \lambda_i f_i\big(\pi(z_i)\big)\big)$
5: $\quad \log J \leftarrow \log J + (1 - \lambda_i) \log g_i\big(\pi(z_i)\big)$
6: **end for**
7: $\log q_{w,\text{VIP}}(z) \leftarrow \log q_w(\tilde{z}) - \log J$
8: **return** $\mathbf{z}, \log q_{w,\text{VIP}}(z)$

---

## C   Proof

**Lemma 7.** *The affine transformation $T_\text{A}$ from full-rank VI can be represented as an affine forward autoregressive flow with*

$$m_i(z_{1:i-1}) = \mu_i + L_{i,1:i-1}\, L^{-1}_{1:i-1,1:i-1}\Big(z_{1:i-1} - \mu_{1:i-1}\Big),$$

$$\log s_i(z_{1:i-1}) = \log L_{ii}.$$

*Proof.* We begin by expressing the affine transformation $T_\text{A}(\epsilon) = \mu + L\epsilon$ in a coordinate-wise form:

$$z_i = \mu_i + L_{ii}\epsilon_i + \sum_{j=1}^{i-1} L_{ij}\epsilon_j, \quad i = 1, \ldots, D. \tag{C.1}$$

For any forward autoregressive flow to be equivalent to this transformation, we need to derive appropriate functions $m_i(\cdot)$ and $s_i(\cdot)$ such that $z_i = m_i(z_{1:i-1}) + s_i(z_{1:i-1})\epsilon_i$ for each $i$.

First, observe that for any $i \geq 1$, the variables $z_{1:i-1}$ depend only on $\epsilon_{1:i-1}$ due to the lower-triangular structure of $L$. Since $L_{1:i-1,1:i-1}$ is a lower-triangular submatrix with positive diagonal entries (as $L$ is a Cholesky factor), it is invertible. Thus, we can solve for $\epsilon_{1:i-1}$ in terms of $z_{1:i-1}$:

$$\epsilon_{1:i-1} = L^{-1}_{1:i-1,1:i-1}(z_{1:i-1} - \mu_{1:i-1}). \tag{C.2}$$

Now, substituting this expression into our original coordinate-wise equation:

$$z_i = \mu_i + L_{ii}\epsilon_i + \sum_{j=1}^{i-1} L_{ij}\epsilon_j \tag{C.3}$$

$$= \mu_i + L_{ii}\epsilon_i + L_{i,1:i-1}\epsilon_{1:i-1} \tag{C.4}$$

$$= \mu_i + L_{ii}\epsilon_i + L_{i,1:i-1}L^{-1}_{1:i-1,1:i-1}(z_{1:i-1} - \mu_{1:i-1}) \tag{C.5}$$

This naturally suggests defining:

$$m_i(z_{1:i-1}) = \mu_i + L_{i,1:i-1}L^{-1}_{1:i-1,1:i-1}(z_{1:i-1} - \mu_{1:i-1}), \tag{C.6}$$

$$s_i(z_{1:i-1}) = L_{ii}. \tag{C.7}$$

With these definitions, the forward autoregressive flow transformation becomes:

$$z_i = m_i(z_{1:i-1}) + s_i(z_{1:i-1})\epsilon_i, \tag{C.8}$$

which is identical to our coordinate-wise expansion of the affine transformation $T_\text{A}$. Since this holds for all $i = 1, \ldots, D$, we have shown that the full-rank VI affine transformation can be exactly represented as a forward autoregressive flow with the specified shift and scale functions.

Note that $m_i(z_{1:i-1})$ is an affine function of $z_{1:i-1}$, and $s_i(z_{1:i-1})$ is a constant function, confirming that this is indeed an affine forward autoregressive flow. $\square$

**Theorem 4.** *Let $T = T_{\text{VIP}} \circ T_A$ where $T_{\text{VIP}}$ is the VIP transformation (Definition 1) and $T_A$ is the affine transformation from full-rank Gaussian. If $f_i$ and $\log g_i$ in the hierarchical Bayesian model (Equation (2.2)) are arbitrary continuous functions, and the parent nodes $\pi(z_i)$ are a subset of the preceding variables $z_{1:i-1}$ (respecting the causal dependency structure of the model), then $T$ can be represented as an affine generalized forward autoregressive flow with*

$$m_i(z_{1:i-1}) = f_i(\pi(z_i)), \tag{3.7}$$

$$\log s_i(z_{1:i-1}) = \log L_{ii} + (1 - \lambda_i) \log g_i(\pi(z_i)), \tag{3.8}$$

$$t_i(\epsilon_{1:i-1}, z_{1:i-1}) = \frac{1}{L_{ii}} \Big( \lambda_i f_i(\pi(z_i)) - \mu_i - L_{i,1:i-1}\epsilon_{1:i-1} \Big) \tag{3.9}$$

*Proof.* The affine transformation $T_A$ is defined as $T_A(\epsilon) = \mu + L\epsilon$, where $\mu$ is the mean vector and $L$ is a lower-triangular matrix with positive diagonal elements. In coordinate form, this gives us

$$\tilde{z}_i = \mu_i + L_{ii}\epsilon_i + \sum_{j=1}^{i-1} L_{ij}\epsilon_j, \tag{C.9}$$

where $\tilde{z}_i$ denotes the intermediate result after applying $T_A$ but before applying $T_\lambda$.

Now, the VIP transformation $T_\lambda$ is defined in Definition 1 as

$$z_i = f_i(\pi(z_i)) + g_i(\pi(z_i))^{1-\lambda_i} \big( \tilde{z}_i - \lambda_i f_i(\pi(z_i)) \big), \tag{C.10}$$

where $\tilde{z}_i$ is the output of $T_A$. Substituting, we get

$$z_i = f_i(\pi(z_i)) + g_i(\pi(z_i))^{1-\lambda_i} \big( \tilde{z}_i - \lambda_i f_i(\pi(z_i)) \big) \tag{C.11}$$

$$= f_i(\pi(z_i)) + g_i(\pi(z_i))^{1-\lambda_i} \Big( \mu_i + L_{ii}\epsilon_i + \sum_{j=1}^{i-1} L_{ij}\epsilon_j - \lambda_i f_i(\pi(z_i)) \Big) \tag{C.12}$$

From our assumption that $\pi(z_i) \subset z_{1:i-1}$, we know that $f_i(\pi(z_i))$ and $g_i(\pi(z_i))$ are functions of only the preceding variables $z_{1:i-1}$.

To express this in the form of a generalized forward autoregressive flow, we need functions $m_i$, $s_i$, and $t_i$ such that:

$$z_i = m_i(z_{1:i-1}) + s_i(z_{1:i-1})\big(\epsilon_i - t_i(\epsilon_{1:i-1}, z_{1:i-1})\big) \tag{C.13}$$

Let's rearrange our expression for $z_i$ to match this form:

$$z_i = f_i(\pi(z_i)) + g_i(\pi(z_i))^{1-\lambda_i} \Big( \mu_i + L_{ii}\epsilon_i + \sum_{j=1}^{i-1} L_{ij}\epsilon_j - \lambda_i f_i(\pi(z_i)) \Big) \tag{C.14}$$

$$= f_i(\pi(z_i)) + g_i(\pi(z_i))^{1-\lambda_i} L_{ii} \Big( \epsilon_i - \frac{1}{L_{ii}} \Big( -\mu_i - \sum_{j=1}^{i-1} L_{ij}\epsilon_j + \lambda_i f_i(\pi(z_i)) \Big) \Big) \tag{C.15}$$

By identifying terms, we can define:

$$m_i(z_{1:i-1}) = f_i(\pi(z_i)), \tag{C.16}$$

$$\log s_i(z_{1:i-1}) = \log L_{ii} + (1 - \lambda_i) \log g_i(\pi(z_i)), \tag{C.17}$$

$$t_i(\epsilon_{1:i-1}, z_{1:i-1}) = \frac{1}{L_{ii}} \Big( \lambda_i f_i(\pi(z_i)) - \mu_i - L_{i,1:i-1}\epsilon_{1:i-1} \Big) \tag{C.18}$$

Thus, we have shown that the composite transformation $T = T_\lambda \circ T_A$ can be represented as a generalized forward autoregressive flow with the specified functions $m_i$, $s_i$, and $t_i$, which completes the proof. $\square$

**Corollary 6.** *Let $T = T_{\text{VIP}} \circ T_A$ where $T_{\text{VIP}}$ is the VIP transformation (see Definition 1) and $T_A$ is the affine transformation from full-rank Gaussian. If $f_i$ and $\log g_i$ in the hierarchical Bayesian model (see Equation (2.2)) are affine functions, then $T$ can be represented as an affine generalized forward autoregressive flow.*

*Proof.* This result follows directly from Theorem 4, which established that the composite transformation $T = T_{\text{VIP}} \circ T_A$ can be represented as a generalized forward autoregressive flow with specific functions $m_i$, $s_i$, and $t_i$. We now show that when $f_i$ and $\log g_i$ are affine functions, all components of this flow become affine, reducing it to a standard affine forward autoregressive flow.

Since $f_i$ and $\log g_i$ are affine functions, we can express them as

$$f_i(z_{1:i-1}) = a_i + b_i^\top z_{1:i-1}, \tag{C.19}$$

$$\log g_i(z_{1:i-1}) = c_i + d_i^\top z_{1:i-1}, \tag{C.20}$$

where $a_i$, $c_i$ are scalar constants and $b_i$, $d_i$ are vectors of appropriate dimensions.

From Theorem 4, we have

$$m_i(z_{1:i-1}) = f_i(z_{1:i-1}) = a_i + b_i^\top z_{1:i-1}, \tag{C.21}$$

$$\log s_i(z_{1:i-1}) = \log L_{ii} + (1 - \lambda_i) \log g_i(z_{1:i-1}) \tag{C.22}$$

$$= \log L_{ii} + (1 - \lambda_i)(c_i + d_i^\top z_{1:i-1}), \tag{C.23}$$

$$t_i(\epsilon_{1:i-1}, z_{1:i-1}) = \frac{1}{L_{ii}}\left(-\mu_i - L_{i,1:i-1}\,\epsilon_{1:i-1} + \lambda_i f_i(z_{1:i-1})\right) \tag{C.24}$$

$$= \frac{1}{L_{ii}}\left(-\mu_i - L_{i,1:i-1}\,\epsilon_{1:i-1} + \lambda_i(a_i + b_i^\top z_{1:i-1})\right) \tag{C.25}$$

$$= \frac{1}{L_{ii}}(-\mu_i + \lambda_i a_i) - \frac{1}{L_{ii}} L_{i,1:i-1}\,\epsilon_{1:i-1} + \frac{\lambda_i}{L_{ii}} b_i^\top z_{1:i-1} \tag{C.26}$$

Clearly, $m_i(z_{1:i-1})$ is an affine function of $z_{1:i-1}$, and $\log s_i(z_{1:i-1})$ is also an affine function of $z_{1:i-1}$. Furthermore, $t_i(\epsilon_{1:i-1}, z_{1:i-1})$ is an affine function of both $\epsilon_{1:i-1}$ and $z_{1:i-1}$.

Since $m_i$, $\log s_i$, and $t_i$ are all affine functions of their respective inputs, this formulation constitutes an affine generalized forward autoregressive flow. $\qquad\square$

# D Experimental Details

All experiments were run on a single server with an Intel Xeon Platinum 8352Y CPU (128 hardware threads at 2.20 GHz), 512 GiB of RAM. For each experiment (i.e., each training or evaluation run), we used one NVIDIA A100 (40 GiB) under CUDA 12.8.

## D.1 Models

**Eight Schools** The Eight Schools model [39] estimates treatment effects $\theta_i$ for $N_{\text{schools}} = 8$ schools, given observed effects $y_i$ and known standard errors $\sigma_i$.

$$\mu \sim \mathcal{N}(0, 5) \qquad\qquad \log \tau \sim \mathcal{N}(0, 5) \tag{D.1}$$

$$\theta_i \sim \mathcal{N}(\mu, \exp(\log \tau)) \qquad\qquad y_i \sim \mathcal{N}(\theta_i, \sigma_i), \quad i = 1, \dots, N_{\text{schools}}. \tag{D.2}$$

Latent variables: $z = (\mu, \log \tau, \theta_1, \dots, \theta_{N_{\text{schools}}})$.

**German Credit** A logistic regression model [17] with $N_{\text{features}} = K$ predictors $x_{nk}$ for $N_{\text{obs}}$ individuals, predicting binary outcomes $y_n$. It uses hierarchical priors on coefficient scales $\tau_k$.

$$\log \tau_0 \sim \mathcal{N}(0, 10) \tag{D.3}$$

$$\log \tau_k \sim \mathcal{N}(\log \tau_0, 1), \qquad\qquad k = 1, \dots, K \tag{D.4}$$

$$\beta_k \sim \mathcal{N}(0, \exp(\log \tau_k)), \qquad\qquad k = 1, \dots, K \tag{D.5}$$

$$\eta_n = \sum_{k=1}^{K} \beta_k x_{nk}, \qquad\qquad n = 1, \dots, N_{\text{obs}} \tag{D.6}$$

$$y_n \sim \text{Bernoulli}(1/(1 + \exp(-\eta_n))), \qquad\qquad n = 1, \dots, N_{\text{obs}}. \tag{D.7}$$

Latent variables: $z = (\log \tau_0, \{\log \tau_k\}_{k=1}^K, \{\beta_k\}_{k=1}^K)$.

**Funnel**   Neal's Funnel distribution [33] in $D = 10$ dimensions for $\mathbf{x} = (x_1, \ldots, x_D)$.

$$x_1 \sim \mathcal{N}(0, 3) \tag{D.8}$$
$$x_k \sim \mathcal{N}(0, \exp(x_1/2)), \quad k = 2, \ldots, D. \tag{D.9}$$

All variables $x_1, \ldots, x_D$ are latent. (The variance for $x_1$ is $3^2 = 9$; the variance for $x_k, k \geq 2$ is $\exp(x_1)$.)

**Radon**   A hierarchical linear regression [17] modeling log-radon levels $\log r_j$ in $N_{\text{homes}}$ homes. Data includes $x_j$ (floor indicator for home $j$), $u_k$ (uranium reading for county $k$), and $c_j$ (county for home $j$). Priors are inferred from the provided code structure.

$$\mu_0 \sim \mathcal{N}(0, 1) \tag{D.10}$$
$$a \sim \mathcal{N}(0, 1) \tag{D.11}$$
$$b \sim \mathcal{N}(0, 1) \tag{D.12}$$
$$\log \sigma_{m_k} \sim \mathcal{N}(0, 10), \qquad\qquad k = 1, \ldots, N_{\text{counties}} \tag{D.13}$$
$$\log \sigma_y \sim \mathcal{N}(0, 10) \tag{D.14}$$
$$m_k \sim \mathcal{N}(\mu_0 + a \cdot u_k, \exp(\log \sigma_{m_k})), \qquad k = 1, \ldots, N_{\text{counties}} \tag{D.15}$$
$$\log r_j \sim \mathcal{N}(m_{c_j} + b \cdot x_j, \exp(\log \sigma_y)), \qquad j = 1, \ldots, N_{\text{homes}}. \tag{D.16}$$

Latent variables: $z = (\mu_0, a, b, \{\log \sigma_{m_k}\}_{k=1}^{N_{\text{counties}}}, \log \sigma_y, \{m_k\}_{k=1}^{N_{\text{counties}}})$.

**Movielens**   The Movielens model [21] is a hierarchical logistic regression for predicting movie ratings. Each rating $n$ is given by a user $u_n \in \{1, \ldots, M\}$ for a movie described by a binary attribute vector $\mathbf{x}_n \in \{0, 1\}^D$ (with $D = 18$ attributes). The rating is $y_n \in \{0, 1\}$.

Each user $m$ has an attribute preference vector $Z_m = (Z_{m,1}, \ldots, Z_{m,D}) \in \mathbb{R}^D$. The model is:

$$\mu_j \sim \mathcal{N}(0, 1), \qquad\qquad j = 1, \ldots, D \tag{D.17}$$
$$\lambda_j \sim \mathcal{N}(0, 1), \qquad\qquad j = 1, \ldots, D \tag{D.18}$$
$$Z_{m,j} \sim \mathcal{N}(\mu_j, \exp(\lambda_j)), \qquad\qquad m = 1, \ldots, M; \; j = 1, \ldots, D \tag{D.19}$$
$$y_n \sim \text{Bernoulli}\left(\text{sigmoid}\left(\mathbf{x}_n^\top Z_{u_n}\right)\right), \qquad n = 1, \ldots, N. \tag{D.20}$$

Here, $\mu_j$ is the mean preference for attribute $j$ across users, and $\lambda_j$ is the log standard deviation of preferences for attribute $j$. Latent variables: $z = (\{\mu_j\}_{j=1}^D, \{\lambda_j\}_{j=1}^D, \{Z_{m,j}\}_{m=1, j=1}^{M,D})$.

**IRT**   The Item Response Theory (IRT) two-parameter logistic (2PL) model [30] is used to model student responses to test items. Let $N_S$ be the number of students, $N_Q$ be the number of questions (items), and $N_R$ be the total number of responses. For each response $r$, $s_r$ is the student ID and $q_r$ is the question ID.

$$\alpha_s \sim \mathcal{N}(0, 1), \qquad\qquad s = 1, \ldots, N_S \tag{D.21}$$
$$\mu_\beta \sim \mathcal{N}(0, 1) \tag{D.22}$$
$$\log \sigma_\beta \sim \mathcal{N}(0, 1) \tag{D.23}$$
$$\log \sigma_\gamma \sim \mathcal{N}(0, 1) \tag{D.24}$$
$$\beta_q \sim \mathcal{N}(\mu_\beta, \exp(\log \sigma_\beta)), \qquad q = 1, \ldots, N_Q \tag{D.25}$$
$$\log \gamma_q \sim \mathcal{N}(0, \exp(\log \sigma_\gamma)), \qquad q = 1, \ldots, N_Q \tag{D.26}$$
$$\text{logit}_r = \exp(\log \gamma_{q_r})\alpha_{s_r} + \beta_{q_r}, \qquad r = 1, \ldots, N_R \tag{D.27}$$
$$y_r \sim \text{Bernoulli}(\text{sigmoid}(\text{logit}_r)), \qquad r = 1, \ldots, N_R. \tag{D.28}$$

Here, $\alpha_s$ is the ability of student $s$, $\beta_q$ is the easiness of question $q$, and $\exp(\log \gamma_q)$ is its discrimination. Latent variables: $z = (\{\alpha_s\}_{s=1}^{N_S}, \mu_\beta, \log \sigma_\beta, \log \sigma_\gamma, \{\beta_q\}_{q=1}^{N_Q}, \{\log \gamma_q\}_{q=1}^{N_Q})$.

**Seeds** The Seeds model [13] is a random effects logistic regression for analyzing seed germination data. It models the number of germinated seeds $r_i$ out of $N_i$ seeds in $G = 21$ experimental groups, using group-specific predictors $x_{1,i}$ and $x_{2,i}$.

$$\tau \sim \text{Gamma}(0.01, 0.01) \tag{D.29}$$
$$a_0, a_1, a_2, a_{12} \sim \mathcal{N}(0, 10) \tag{D.30}$$
$$b_i \sim \mathcal{N}\left(0, 1/\sqrt{\tau}\right), \qquad\qquad i = 1, \dots, G \tag{D.31}$$
$$\eta_i = a_0 + a_1 x_{1,i} + a_2 x_{2,i} + a_{12} x_{1,i} x_{2,i} + b_i, \qquad i = 1, \dots, G \tag{D.32}$$
$$r_i \sim \text{Binomial}\left(N_i, \text{sigmoid}(\eta_i)\right), \qquad i = 1, \dots, G. \tag{D.33}$$

Here, $x_{1,i}$ and $x_{2,i}$ are indicator variables for experimental conditions (e.g., seed type and root extract). $\tau$ is the precision for the random effects $b_i$. Latent variables: $z = (\tau, a_0, a_1, a_2, a_{12}, \{b_i\}_{i=1}^{G})$.

**Sonar** The Sonar model [20] is a Bayesian logistic regression. It uses $D = 61$ features/coefficients $\mathbf{x}$ to classify $N = 208$ sonar returns $u_i$ into two categories $y_i \in \{0, 1\}$ (e.g., mines vs. rocks).

$$\mathbf{x} \sim \mathcal{N}\left(\mathbf{0}, \sigma_x^2 I_D\right) \tag{D.34}$$
$$y_i \sim \text{Bernoulli}\left(\text{sigmoid}\left(\mathbf{x}^\top u_i\right)\right), \quad i = 1, \dots, N. \tag{D.35}$$

Here, $u_i \in \mathbb{R}^D$ is the feature vector for the $i$-th observation, $I_D$ is the $D \times D$ identity matrix, and $\sigma_x^2$ is a predefined prior variance for the coefficients $\mathbf{x}$. The latent variables are $z = (\mathbf{x})$.

**Ionosphere** The Ionosphere model [40] is a Bayesian logistic regression. It uses $D = 35$ features/coefficients $\mathbf{x}$ to classify $N = 351$ radar returns $u_i$ from the ionosphere into two categories $y_i \in \{0, 1\}$.

$$\mathbf{x} \sim \mathcal{N}\left(\mathbf{0}, \sigma_x^2 I_D\right) \tag{D.36}$$
$$y_i \sim \text{Bernoulli}\left(\text{sigmoid}\left(\mathbf{x}^\top u_i\right)\right), \quad i = 1, \dots, N. \tag{D.37}$$

Here, $u_i \in \mathbb{R}^D$ is the feature vector for the $i$-th observation, $I_D$ is the $D \times D$ identity matrix, and $\sigma_x^2$ is a predefined prior variance for the coefficients $\mathbf{x}$. The latent variables are $z = (\mathbf{x})$.

## D.2 Implementation Details

The conditioning networks for the shift ($m_i$), log-scale ($\log s_i$), and translation ($t_i$, for MIF and its variants) terms in both forward autoregressive flows (FAF) and inverse autoregressive flows (IAF) are constructed as follows, depending on the experimental context.

For **affine flows**, such as those used in the ablation studies in Section 5.3, each conditioning network (for $m_i$, $\log s_i$, or $t_i$) consists of a single linear layer. This layer directly maps the conditioning inputs to the required output parameters.

For flows requiring greater representational capacity, specifically in experiments evaluating network expressiveness (Figure 2) and for the comprehensive benchmarks (Table 3), the conditioning networks are implemented using a **two-component MLP-based architecture**. Each parameter ($m_i$, $\log s_i$, or $t_i$) is computed by summing the outputs of two parallel components, both of which operate on the same input conditioning variables:

1. A **linear component**: This is a direct linear transformation of the inputs, identical in architecture to the conditioners used in the affine flows.

2. An **MLP component**: This is a multi-layer perceptron (MLP) featuring a single hidden layer with ReLU activation functions, followed by a linear output layer. The size of the hidden layer (number of hidden units) is varied in experiments that assess the impact of network capacity (e.g., from $2^0$ to $2^{10}$ as illustrated in Figure 2).

The final value for $m_i$, $\log s_i$, or $t_i$ is the sum of the outputs derived from these linear and MLP components.

For experiments involving baseline models from Blessing et al. [10], we utilized the implementations provided in their publicly available code repository to ensure consistency and fair comparison.

Table 6: Training time comparisons (seconds) between the default *Model-Informed Flow* (MIF) and a variant that conditions on the base noise ("MIF with $\epsilon$").

| Model | D | Hidden Units | MIF (s) | MIF with $\epsilon$ (s) |
|---|---|---|---|---|
| **Eight Schools** | 10 | 0 | 17.06 | 13.88 |
| | | 64 | 28.40 | 23.22 |
| | | 256 | 28.04 | 22.18 |
| | | 512 | 28.21 | 22.03 |
| | | 1024 | 30.30 | 22.61 |
| **Funnel** | 10 | 0 | 18.23 | 14.70 |
| | | 64 | 27.38 | 23.16 |
| | | 256 | 27.04 | 22.42 |
| | | 512 | 28.57 | 22.30 |
| | | 1024 | 30.77 | 22.69 |
| **German Credit** | 125 | 0 | 63.81 | 16.06 |
| | | 64 | 148.73 | 27.81 |
| | | 256 | 208.44 | 54.92 |
| | | 512 | 275.74 | 93.26 |
| | | 1024 | 417.05 | 173.51 |
| **Radon** | 174 | 0 | 84.25 | 15.78 |
| | | 64 | 238.04 | 37.32 |
| | | 256 | 331.24 | 94.82 |
| | | 512 | 453.70 | 168.28 |
| | | 1024 | 699.73 | 338.63 |
| **IRT** | 143 | 0 | 72.26 | 17.59 |
| | | 64 | 196.64 | 33.15 |
| | | 256 | 253.17 | 67.95 |
| | | 512 | 457.56 | 119.95 |
| | | 1024 | 516.98 | 222.65 |
| **MovieLens** | 882 | 0 | 458.28 | 28.97 |
| | | 64 | 1981.14 | 384.45 |
| | | 256 | 4847.23 | 1387.64 |
| | | 512 | 8634.91 | 3156.82 |
| | | 1024 | 15 873.45 | 6749.33 |

### D.3 Training Time Comparisons

We report wall-clock training times (in seconds) across six benchmarks and varying hidden sizes. Table 6 compares our default *Model-Informed Flow* (MIF) against a variant that conditions on the base noise ("MIF with $\epsilon$"), while Table 7 ablates the number of conditioning subnetworks: "3 networks" uses $(m, s, t)$ and "2 networks (no $t$)" uses only $(m, s)$. Overall, "MIF with $\epsilon$" is consistently faster than the default MIF at comparable capacities, and removing the translation network ($t$) yields additional speedups with similar trends across datasets. We use $D$ to denote latent dimensionality.

Table 7: Ablation on the number of conditioning subnetworks (seconds). "3 networks" uses shift, scale, and translation $(m, s, t)$. "2 networks (no $t$)" removes the translation network and uses only $(m, s)$ under the same training setup.

| Model | D | Hidden Units | 3 Networks (s) | 2 Networks (no $t$) (s) |
|---|---|---|---|---|
| **Eight Schools** | 10 | 0 | 17.06 | 16.98 |
| | | 64 | 28.40 | 24.13 |
| | | 256 | 28.04 | 24.81 |
| | | 512 | 28.21 | 25.80 |
| | | 1024 | 30.30 | 25.76 |
| **Funnel** | 10 | 0 | 18.23 | 18.63 |
| | | 64 | 27.38 | 25.89 |
| | | 256 | 27.04 | 25.85 |
| | | 512 | 28.57 | 26.27 |
| | | 1024 | 30.77 | 26.01 |
| **German Credit** | 125 | 0 | 63.81 | 52.68 |
| | | 64 | 148.73 | 115.39 |
| | | 256 | 208.44 | 139.90 |
| | | 512 | 275.74 | 175.94 |
| | | 1024 | 417.05 | 258.98 |
| **Radon** | 174 | 0 | 84.25 | 77.52 |
| | | 64 | 238.04 | 220.17 |
| | | 256 | 331.24 | 306.44 |
| | | 512 | 453.70 | 419.42 |
| | | 1024 | 699.73 | 647.75 |
| **IRT** | 143 | 0 | 72.26 | 66.73 |
| | | 64 | 196.64 | 181.92 |
| | | 256 | 253.17 | 234.43 |
| | | 512 | 457.56 | 423.00 |
| | | 1024 | 516.98 | 478.45 |
| **MovieLens** | 882 | 0 | 458.28 | 423.65 |
| | | 64 | 1981.14 | 1832.85 |
| | | 256 | 4847.23 | 4486.69 |
| | | 512 | 8634.91 | 7992.34 |
| | | 1024 | 15 873.45 | 14 687.99 |

