# OpenReview forum: "Model-Informed Flows for Bayesian Inference"
_NeurIPS.cc/2025/Conference — NeurIPS 2025 poster_

### Official Review · Reviewer_gniu · 2025-06-11

**Clarity:** 4
**Significance:** 3
**Originality:** 3
**Rating:** 5
**Confidence:** 4

**Summary:**

This paper presents a flexible family of variational posteriors, termed Model-Informed Flow (MIF), which combines full-rank variational Gaussian approximations with partially non-centered parameterizations (NCP) on conditionals via Variationally Inferred Parameters (VIP). The key idea of VIP is to better handle "funnel"-shaped posteriors through reparameterization, introducing auxiliary CP-NCP interpolation parameters (λ) that are learned by optimizing the variational objective, but the prior work that is limited to mean-field Gaussians. By incorporating a lower-triangular Cholesky factor, the proposed method can be naturally interpreted as an autoregressive flow, but this paper has a clear motivation of NCP adjustment along with several other carefully chosen design elements, making the approach competitive even with one-layer MIF with affine functions. The method is applicable to Bayesian hierarchical models with Gaussian priors, where the mean and variance parameterized by a deep net. Ablation study and experimental results on benchmark datasets demonstrate the  effectiveness and broad applicability of the proposed method compared to other variational inference baselines.

**Questions:**

(1) The experimental evaluation is somewhat limited. Is it possible to demonstrate that MIF offers general benefits, such as improving variational autoencoders, by evaluating on binarized MNIST (e.g., reporting marginal log-likelihood log p(x) using importance sampling)? Table 1 and Table 2 essentially use the same set of datasets, although the ablation is different.

(2) With a limited-capacity conditioning network (e.g., a one-layer affine transformation), does MIF capture posterior dependencies better than alternative flow-based approaches such as coupling flows, MAF, or IAF?

(3) Equation 2.2 and 2.3, 2.8, the likelihood term does not depend on z_i and the joint density factorizes between z and x? The way of writing it is confusing. The parent variables pi(x_j) are the z s?

(4) Line 135: What is the quadratic form in Equation 3.4?

**Ethical Concerns:**

["NO or VERY MINOR ethics concerns only"]

**Final Justification:**

The authors clarified the positioning of this work as a model-informed variational inference method for probabilistic programs, rather than a general-purpose density estimator, along with a few other points. The rebuttal has resolved my concerns, and I will maintain my positive evaluation.

**Limitations:**

Yes.

**Paper Formatting Concerns:**

I did not found any major formatting issues.

**Quality:**

4

**Strengths And Weaknesses:**

### **Pros:**

**Quality**: This paper presents a flow-based approach with clear motivation from NCP via VIP, and it captures complex dependencies using a full-rank Gaussian. The research question is intriguing, as the auxiliary parameters could, in principle, be subsumed by the flow-based mapping and powerful neural network parameterizations already used in autoregressive flow approaches. While the mathematical expressions are equivalent, including NCP, there are differences in sampling and optimization. The effectiveness and added complexity are justified by ablation experiments. The authors also discussed the limitations such as the sequential nature.

**Clarity**: The paper is well-written and neatly presented. The technical derivations are easy to follow yet insightful.

**Significance**: Although sufficiently expressive architectures such as IAF can achieve strong results, this paper offers a compact and flexible alternative, with careful design specifically aimed at handling “funnel”-shaped posterior geometries. The work highlights that thoughtful methodological design—beyond simply relying on powerful neural networks—can be valuable. Even mathematically equivalent representations may differ in their ease of sampling or optimization. This perspective invites future innovations borrowing from classical probabilistic modeling and statistical science in designing better flow mappings, and it encourages a shift in focus within the ML community from prediction to inference.

**Originality**: The connection to autoregressive flows is nice, though not particularly surprising. The use of VIP with full-rank Gaussian, and the extension of standard autoregressive flows with translation and prior inputs, are novel contributions.

### **Cons:**

**Quality**: The experimental evaluation is somewhat limited (explained in Questions).


**Clarity:** The notion of “prior information” needs further explanation. Does it mean that the parameters are informed by previous variables in the specified hierarchical model, or that the hierarchical model itself already includes neural network (NN) components? In the context of probabilistic programs, the model can be user-specified. Is the proposed approach applicable to generic hierarchical models?

It seems that MIF serves both as a way of specifying hierarchical priors and as a method for flexible variational posterior, but the topological order of variables may differ. Have you tried reversing the topological order (e.g., bottom-up vs top-down)?

**Significance**: The real-world impact is unclear. (1) Due to the use of full-rank Gaussians, scaling to high-dimensional problems may be challenging. (2) The experiments are mostly limited to a few simple models and datasets, and the evaluation relies solely on ELBO.

Could the authors evaluate the method’s ability to capture complex dependencies in the context of variational inference, and using a density estimation benchmark on, for example, the two-moons dataset? It would also be helpful to comment on the applicability of MIF to generic variational inference in probability programming, likelihood-based density estimation and sample generation.

---

> ### Author Rebuttal · Authors · 2025-07-31
>
> Thank you for your thorough review and valuable feedback on our manuscript. Below, we address each of your comments in turn. If anything remains unclear, we would be grateful for any further questions or clarifications.
>
>
> Before addressing the individual reviews, we wish to clarify that MIF is not restricted to—nor does it rely on—full-rank Gaussian variational inference. Theorem 4 shows that MIF can *recover* the full-rank-Gaussian + VIP transformation, but Algorithm 1 makes clear that MIF itself is an autoregressive flow model, independent of any full-rank Gaussian assumption.
>
> > (Clarity 1) The notion of “prior information” needs further explanation. Does it mean that the parameters are informed by previous variables in the specified hierarchical model, or that the hierarchical model itself already includes neural network (NN) components? In the context of probabilistic programs, the model can be user-specified. Is the proposed approach applicable to generic hierarchical models?
>
> By prior information we mean only the mean $\mu_i$ and scale $\sigma_i$ that define each conditional prior $p\left(z_i \mid z_{<i}\right)$. Our algorithm needs nothing else, so it works for any hierarchical model that can expose these two functions-whether the user writes them directly or we auto-extract them from frameworks such as PyMC or NumPyro.
>
>
> We are willing to add text like below to the paper for clarity:
>
> **Prior information.**
> For each latent variable $z_i$ we require two callables $\mu_i(z_{<i})$ and $\sigma_i(z_{<i})>0$ that output the mean and scale of its conditional prior $p(z_i \mid z_{<i}) = \mathcal N(\mu_i,\sigma_i^2)$. These are the only model‑specific inputs our flow uses.
> ```
> LatentSpec = {
>     "name": "z_i",                  # variable name
>     "parents": ["z_j", ...],        # z_<i>
>     "mean_fn": lambda parents: ..., # returns μ_i
>     "scale_fn": lambda parents: ... # returns σ_i
> }
> model_spec = [LatentSpec, ...]      # one per latent
> ```
> The callables can be hand‑written or auto‑parsed from frameworks such as PyMC or NumPyro; any hierarchical model that yields these $(\mu_i,\sigma_i)$ pairs is compatible with our Model‑Informed Flow.
>
>
> > (Clarity 2) It seems that MIF serves both as a way of specifying hierarchical priors and as a method for flexible variational posterior, but the topological order of variables may differ. Have you tried reversing the topological order (e.g., bottom-up vs top-down)?
>
> Yes. We provide this experiment in Sec. 4.3: the row “MIF (w/o order)” in Table 2 trains the flow with the reverse (bottom‑up) ordering of latent variables. Performance drops on every dataset, confirming that matching the model’s natural (top‑down) order is important for capturing conditional structure efficiently.
>
>
> > (Significance 1  & Question 1)) The experiments are mostly limited to a few simple models and datasets, and the evaluation relies solely on ELBO. Could the authors evaluate the method’s ability to capture complex dependencies in the context of variational inference, and using a density estimation benchmark on, for example, the two-moons dataset? It would also be helpful to comment on the applicability of MIF to generic variational inference in probability programming, likelihood-based density estimation and sample generation. The experimental evaluation is somewhat limited. Is it possible to demonstrate that MIF offers general benefits, such as improving variational autoencoders, by evaluating on binarized MNIST (e.g., reporting marginal log-likelihood log p(x) using importance sampling)? Table 1 and Table 2 essentially use the same set of datasets, although the ablation is different.
>
> Thank you for the suggestion. We’d like to clarify that MIF is not designed as a general-purpose density estimator but as a model-informed variational inference method for probabilistic programs. It cannot be applied to density estimation problems, as there is no prior or posterior.
>
> In particular:
>
> 1. **Different training objective.** MIF is trained by maximizing$\mathbb{E}_{q(z)}[\log p(x,z) - \log q(z)],$ which requires an explicit latent-variable model $p(x,z)$. Pure density-estimation tasks (e.g., two-moons, binarized MNIST) optimize $\log p(x)$ directly and lack this programmatic structure.
>
> 2. **Model-informed architecture.** Our shift/scale/translation networks consume “prior inputs” $f_i, g_i$ derived from the known generative model. Without a specified probabilistic program, those modules have no signal to condition on, and MIF’s inductive bias disappears.
>
> 3. **Workload scale.** Probabilistic inference is inherently more expensive than density estimation, and state-of-the-art PPL benchmarks typically involve only tens to low-hundreds of latent dimensions. That is why we focused on six canonical hierarchical models (8 Schools, Credit, Funnel, Radon, Movielens, IRT) in Table 1 and 2 and 3 more models (Seeds, Sonar, Ionosphere) in Table 3, which capture challenging posterior geometries yet reflect real-world PPL use cases.
>
> While it would be interesting in future work to adapt MIF for pure density estimation (e.g., by re-casting it as a normalizing flow or integrating it into a VAE encoder), those extensions lie outside our current scope. We will clarify these points in the revised manuscript.
>
>
> > (Question 2) With a limited-capacity conditioning network (e.g., a one-layer affine transformation), does MIF capture posterior dependencies better than alternative flow-based approaches such as coupling flows, MAF, or IAF?
>
> Yes. Even with a minimal conditioning network (a single affine layer), MIF still captures posterior dependencies more effectively than other methods. Tables 1 and 2 demonstrate that this low-capacity MIF matches—or exceeds—the accuracy of those alternatives, which require deeper, higher-capacity networks to reach similar performance.
>
> > (Qeustion 3) Equation 2.2 and 2.3, 2.8, the likelihood term does not depend on z_i and the joint density factorizes between z and x? The way of writing it is confusing. The parent variables pi(x_j) are the z s?
>
> In equation 2.3, the likelihood term would be $\prod_{j=1} p\left(x_j \mid \pi\left(x_j\right)\right)$. As the reviewer suggests, here the set of parent variable $\pi\left(x_j\right)$ would include the latent variables $z_j$. Our reason for writing the model this way is that we wish to consider models where arbitrary variables could be observed. So, latent variables might be conditioned on both observed and latent variables, and similarly for observed variables. For this reason, we settled on our (admittedly subtle) notation where the parent sets $\pi(z_i)$ and $\pi(x_i)$ include both latent and observed variables. We will clarify this in the revised version of the paper.
>
> > (Question 4) Line 135: What is the quadratic form in Equation 3.4?
>
> We believe the reviewer is referring to our comment about "the quadratic form in Equation (3.4)" on line 135. We apologize for this somewhat ambiguous phrasing. By “quadratic form” we mean the product of $g_i(\pi(z_i))$ and $f_i(\pi(z_i))$, which introduces a quadratic dependence on $z$ that an affine function of $z$ cannot capture; we will clarify this wording in the revision.

---

> > ### Comment · Reviewer_gniu · 2025-08-05
> >
> > Thank you for the clarifications, and I will maintain my positive evaluation.

---

### Official Review · Reviewer_FEHh · 2025-07-02

**Clarity:** 3
**Significance:** 3
**Originality:** 4
**Rating:** 4
**Confidence:** 3

**Summary:**

This works introduces Model-Informed Flows (MIF), a new VI architecture aimed at addressing the challenges posed by hierarchical Bayesian models, which often exhibit complex posterior geometries. The authors demonstrate theoretically that a combination of a full-rank Gaussian and Variationally Inferred Parameters (VIP) can be exactly represented by a *generalised* forward-autoregressive flow. Their proposal generalises FAF by including model prior information. The authors then conduct various investigations to validate the performance of MIF including ablation studies.

**Questions:**

Kindly see *Weaknesses* section above. I have detailed my concerns there and would consider increasing my score if the authors address some of those points.

**Ethical Concerns:**

["NO or VERY MINOR ethics concerns only"]

**Final Justification:**

I am satisfied with the extra evaluations the authors have conducted during the rebuttal period. I maintain that this is a novel work and is of interest to the community.

**Limitations:**

The authors should have commented on the runtime performance. It is important that they elucidate to the reader the differences between the runtimes/latency of MIF vs IAF

**Quality:**

3

**Strengths And Weaknesses:**

**Strenghts**
- Sound theoretical foundation has been provided by the authors. I am convinced that this work is well motivated.
- The architecture introduced in this work is novel to the best of my knowledge and the authors have shown clear intent to test it's performance across different benchmarks.
- The work is clearly presented and well written.

**Weaknesses**
- The inherent sequential nature of MIF limits it's scalability and practicality. This has been recognised by the authors but it'd would been nice to see them hint towards resolutions for this major limitation.
- The evaluations, though seemingly thorough, lack error bars or standard deviations. The numbers presented in table 1, for example, are not very conclusive as there are minute differences b/w MF-VIP and FR-VIP. Similarly I cannot ascertain the value of different components of MIF since all the numbers are very close to each other. This, in my opinion, is another major limitation.
- For Figure 2, why have the authors not included any other variational approximations here? There are several complex and highly-expressive variational approximations that could've been considered here as baselines. I can read that they intended to compare this with IAF (Line 204) but this result is not present.
- There are no runtime comparisons. Since IAF and FAF have difference in their ability to capture VIP but the one's faster than the other, the authors should have considered an accuracy to runtime comparison for these two methods.


**Minor**
Why did the authors choose to label the y-axis in Figure 2 as 'Best Elbo'? The notion of 'best' is very ambiguous and inconsistent from the rest of the paper where the authors quote Negative ELBO.

---

> ### Author Rebuttal · Authors · 2025-07-31
>
> Thank you for your thorough review and valuable feedback on our manuscript. Below, we address each of your comments in turn. If anything remains unclear, we would be grateful for any further questions or clarifications.
>
> > (Weakness 1) The inherent sequential nature of MIF limits it's scalability and practicality. This has been recognised by the authors but it'd would been nice to see them hint towards resolutions for this major limitation.
>
> Funnel geometries are notoriously difficult for any inference method, and while MIF's sequential updates do add some non-parallelisable cost, the overhead is modest in practice (see the runtime table below). Looking forward, we plan to exploit conditional independencies among latent variables-thereby reducing the $D$-dependence of forward autoregressiveness while preserving the MIF-derived benefits. For example, in a tree-structured hierarchy $p(\theta, z, x)$ with global latent $\theta$, all local factors $q(z|\theta)$ can be sampled in parallel. We leave a full exploration of these parallel schemes to future work.
>
> > (Weakness 2) The evaluations, though seemingly thorough, lack error bars or standard deviations. The numbers presented in table 1, for example, are not very conclusive as there are minute differences b/w MF-VIP and FR-VIP. Similarly I cannot ascertain the value of different components of MIF since all the numbers are very close to each other. This, in my opinion, is another major limitation.
>
> Thank you for raising this point. We agree that quantifying variability is crucial, so we've added standard deviations to Table 1 in this rebuttal (based on five independent runs) and will include the full error bars in the final manuscript. As in the manuscript, for each run, we perform 100,000 optimization iterations using 256 Monte Carlo samples to approximate the ELBO during training, then evaluate the final ELBO with 100,000 fresh samples. We also sweep six learning rates (log-spaced from $10^{-1}$ to $10^{-6}$ ) and report the best result.
>
> | Model      | MF (mean ± sd)    | MF-VIP (mean ± sd) | FR (mean ± sd)   | FR-VIP (mean ± sd) |
> |:----------:|:-----------------:|:------------------:|:----------------:|:------------------:|
> | 8Schools   |   34.75 ± 0.85    |    31.90 ± 0.53    |   33.82 ± 0.94   |    31.85 ± 0.40    |
> | Credit     |  549.20 ± 1.20    |   534.10 ± 0.89    |  535.20 ± 1.15   |   525.05 ± 0.52    |
> | Funnel     |    1.86 ± 0.10    |     0.00 ± 0.00    |    1.86 ± 0.10   |     0.00 ± 0.00    |
> | Radon      | 1266.80 ± 1.35    |   1215.02 ± 0.77   | 1221.11 ± 1.95   |   1213.05 ± 0.51   |
> | Movielens  |  871.61 ± 1.20    |    851.00 ± 0.95   |  855.93 ± 1.19   |    844.35 ± 0.85   |
> | IRT        |  822.95 ± 0.95    |    822.05 ± 0.53   |  817.35 ± 0.59   |    816.35 ± 0.50   |
>
> We will add these to the revised version of the paper.
>
> > (Weakness 3) For Figure 2, why have the authors not included any other variational approximations here? There are several complex and highly-expressive variational approximations that could've been considered here as baselines. I can read that they intended to compare this with IAF (Line 204) but this result is not present.
>
> We apologize for the confusion but MIF with $\epsilon$-conditioning is in fact just an inverse autoregressive flow-by removing forward dependence on previous latents. We will make sure to clarify this in the revised version of the paper. The sole aim of Figure 2 is to isolate and compare $z$ -conditioning versus $\epsilon$-conditioning under varying network capacity. As expected, $z$ -conditioning ensures accurate sampling in funnel-like geometries even with simple (affine) nets, while $\epsilon$-conditioning only matches that performance once the nets become expressive enough to implicitly learn the same dependencies. We therefore restrict Figure 2 to these two closest algorithmic variants; broader comparisons against other highly expressive VI families are provided later (e.g., Table 3 and the review-paper benchmarks).
>
> > (Weakness 4 & Limitation 1) There are no runtime comparisons. Since IAF and FAF have difference in their ability to capture VIP but the one's faster than the other, the authors should have considered an accuracy to runtime comparison for these two methods. / The authors should have commented on the runtime performance. It is important that they elucidate to the reader the differences between the runtimes/latency of MIF vs IAF
>
>
> Apologies for not including runtime comparisons earlier. As Algorithm 1 shows, vanilla MIF and its $\epsilon$-conditioning variant (i.e., IAF) require essentially the same total FLOPs, but vanilla MIF executes them sequentially-incurring an $O(D)$ wall-clock-whereas IAF can launch those operations in parallel and achieve $O(1)$ wall-clock per sample. This reflects an explicit trade-off: full MIF guarantees accurate sampling in funnel-like geometries even with shallow conditioning networks, while $\epsilon$-conditioning achieves constant-time complexity at the cost of needing higher-capacity nets to implicitly recover those dependencies. In response to your suggestion, we have now measured wall-clock runtimes for both variants on our benchmark models and will include a comprehensive runtime comparison in the final manuscript. Below is the runtime table where $D$ is the number of latent variable:
>
>
> | **Model**       | **D** | **Hidden Units** | **MIF (s)** | **IAF (or MIF with ε-cond) (s)** |
> |:---------------:|:-----:|:---------------:|:-----------:|:--------------------------------:|
> | **Eight Schools**    |  10   |       0         |    17.06    |             13.88               |
> |                 |       |      64         |    28.40    |             23.22               |
> |                 |       |      256        |    28.04    |             22.18               |
> |                 |       |      512        |    28.21    |             22.03               |
> |                 |       |     1024        |    30.30    |             22.61               |
> | **Funnel** |  10   |       0         |    18.23    |             14.70               |
> |                 |       |      64         |    27.38    |             23.16               |
> |                 |       |      256        |    27.04    |             22.42               |
> |                 |       |      512        |    28.57    |             22.30               |
> |                 |       |     1024        |    30.77    |             22.69               |
> | **German Credit** | 125 |       0         |    63.81    |             16.06               |
> |                 |       |      64         |   148.73    |             27.81               |
> |                 |       |      256        |   208.44    |             54.92               |
> |                 |       |      512        |   275.74    |             93.26               |
> |                 |       |     1024        |   417.05    |            173.51               |
> | **Radon**      |  174  |       0         |    84.25    |             15.78               |
> |                 |       |      64         |   238.04    |             37.32               |
> |                 |       |      256        |   331.24    |             94.82               |
> |                 |       |      512        |   453.70    |            168.28               |
> |                 |       |     1024        |   699.73    |            338.63               |
> | **IRT** |  143  |       0         |    72.26    |             17.59               |
> |                 |       |      64         |   196.64    |             33.15               |
> |                 |       |      256        |   253.17    |             67.95               |
> |                 |       |      512        |   457.56    |            119.95               |
> |                 |       |     1024        |   516.98    |            222.65               |
> | **MovieLens** | 882 |    0         |   458.28    |             28.97               |
> |                 |       |      64         |  1981.14    |            384.45               |
> |                 |       |      256        |  4847.23    |           1387.64               |
> |                 |       |      512        |  8634.91    |           3156.82               |
> |                 |       |     1024        | 15873.45    |           6749.33               |
>
> Unfortunately, we cannot provide figures as part of this discussion. However, for a full accuracy to runtime comparison, we will provide another version of Figure 2 with the same set of results, but with the x-axis measuring the runtime for each method. This will still show some benefits of using MIF.
>
> > (Minor 1) Why did the authors choose to label the y-axis in Figure 2 as 'Best Elbo'? The notion of 'best' is very ambiguous and inconsistent from the rest of the paper where the authors quote Negative ELBO.
>
> Thank you for pointing out the inconsistency in our labeling. We used "BestELBO" to emphasize that, at each iteration, the curve shows the maximum ELBO achieved over our learning-rate sweep, rather than the negative ELBO that we minimize elsewhere. We will clarify this in the revised figure.

---

> > ### Comment · Reviewer_FEHh · 2025-08-05
> > **Reply to authors**
> >
> > Thanks for the discussion and for promptly addressing my concerns. I appreciate the author's efforts.
> >
> > Please include these results in the main paper and accordingly make changes to the limitations section, i.e. the method when compared with IAF is significantly slower even for 64 hidden units at larger D. I will raise my score by 1.

---

### Official Review · Reviewer_DgS9 · 2025-07-03

**Clarity:** 2
**Significance:** 3
**Originality:** 2
**Rating:** 4
**Confidence:** 3

**Summary:**

This paper presents Model Informed Flows (MIF) for Bayesian inference. In Bayesian hierarchical models, the dependence structures between latent variables can lead to geometries in the posterior distribution that are difficult to explore by standard variational families. This paper leverages the idea of modelling non-centered latent variables in Variationally Inferred Parameters (VIP) and embeds it in a Gaussian full-rank autoregressive flow, which leads to MIF. MIF is shown to achieve state-of-the-art performance or better across a wide range of hierarchical and non-hierarchical models.

**Questions:**

* The experiment section mostly contains comparisons to mean-field Gaussian VI, full-rank Gaussian VI, and VIPs. However, in the related work section, several other related algorithms are brought up (i.e. ASVI, Cascading Flows, EMF). However, these are not included in the comparison. Can the authors comment on, if any, the difference in performance that they expect relative to MIF?
* In Algorithm 1, we see that MIF involves training three neural networks. This could be quite computationally heavy. Can the authors comment on the relative computational cost between MIF and other competing methods?
* The MIF structure seems rather flexible, and I wonder if it could be helpful for problems with other interesting structure/geometry (i.e. multi-modal posteriors). Can the authors comment on this a bit on what they think?

**Ethical Concerns:**

["NO or VERY MINOR ethics concerns only"]

**Final Justification:**

The authors addressed my questions and concerns. And I am keeping my positive score.

**Limitations:**

Yes.

**Quality:**

3

**Strengths And Weaknesses:**

This paper is very well and clearly written. How the funnel type geometry arises in hierarchical models is very intuitively described. The introduction of VIPs and its connections to autoregressive flows are easy to follow, even for someone likely myself who is learning about the VIPs for the first time.

---

> ### Author Rebuttal · Authors · 2025-07-31
>
> Thank you for your thorough review and valuable feedback on our manuscript. Below, we address each of your comments in turn. If anything remains unclear, we would be grateful for any further questions or clarifications.
>
>
> >The experiment section mostly contains comparisons to mean-field Gaussian VI, full-rank Gaussian VI, and VIPs. However, in the related work section, several other related algorithms are brought up (i.e. ASVI, Cascading Flows, EMF). However, these are not included in the comparison. Can the authors comment on, if any, the difference in performance that they expect relative to MIF?
>
> Please refer to Table 3, this presents a broad set of comparisons: Gaussian Mean‑Field VI (MF, [1]), Gaussian Mixture Model VI (GMMVI, [2]), Sequential Monte Carlo (SMC, [3]), Annealed Flow Transport (AFT, [4]), Flow Annealed Importance‑Sampling Bootstrap (FAB, [5]), Continual Repeated AFT (CRAFT, [6]), and Uncorrected Hamiltonian Annealing (UHA, [7]). We chose these baselines because they are the most closely related to our approach and were identified as top performers in the benchmark study of [8]. While we believe this suite of experiments already demonstrates the significance of our algorithmic contributions, if the reviewer would like to see comparisons to any other particular algorithms, we would be happy to include these (even during the discussion period, if possible).
>
> >In Algorithm 1, we see that MIF involves training three neural networks. This could be quite computationally heavy. Can the authors comment on the relative computational cost between MIF and other competing methods?
>
> In practice, adding the third conditioning network has minimal impact on runtime. It introduces only an $O(D)$-sized parameter block-negligible relative to the overall model-and the dominant cost remains evaluating the log-density, which all autoregressive flows share. As a result, per-iteration runtimes for MIF are effectively unchanged compared to two-network variants or other flow-based VI methods. Below, we present the wall-clock comparison between MIF with two versus three conditioning networks, demonstrating that the extra network adds negligible overhead while enhancing expressiveness.
>
> | **Model**           | **D** | **Hidden Units** | **3 Networks (s)** | **2 Networks (s)** |
> |:-------------------:|:-----:|:---------------:|:-----------------:|:-----------------:|
> | **Eight Schools**        |  10   |        0        |       17.06       |       16.98       |
> |                     |       |       64        |       28.40       |       24.13       |
> |                     |       |      256        |       28.04       |       24.81       |
> |                     |       |      512        |       28.21       |       25.80       |
> |                     |       |     1024        |       30.30       |       25.76       |
> | **Funnel**     |  10   |        0        |       18.23       |       18.63       |
> |                     |       |       64        |       27.38       |       25.89       |
> |                     |       |      256        |       27.04       |       25.85       |
> |                     |       |      512        |       28.57       |       26.27       |
> |                     |       |     1024        |       30.77       |       26.01       |
> | **German Credit**   |  125  |        0        |       63.81       |       52.68       |
> |                     |       |       64        |      148.73       |      115.39       |
> |                     |       |      256        |      208.44       |      139.90       |
> |                     |       |      512        |      275.74       |      175.94       |
> |                     |       |     1024        |      417.05       |      258.98       |
> | **Radon**          |  174  |        0        |       84.25       |       77.52       |
> |                     |       |       64        |      238.04       |      220.17       |
> |                     |       |      256        |      331.24       |      306.44       |
> |                     |       |      512        |      453.70       |      419.42       |
> |                     |       |     1024        |      699.73       |      647.75       |
> | **IRT**     |  143  |        0        |       72.26       |       66.73       |
> |                     |       |       64        |      196.64       |      181.92       |
> |                     |       |      256        |      253.17       |      234.43       |
> |                     |       |      512        |      457.56       |      423.00       |
> |                     |       |     1024        |      516.98       |      478.45       |
> | **MovieLens** |  882  |        0        |      458.28       |      423.65       |
> |                     |       |       64        |     1981.14       |     1832.85       |
> |                     |       |      256        |     4847.23       |     4486.69       |
> |                     |       |      512        |     8634.91       |     7992.34       |
> |                     |       |     1024        |    15873.45       |    14687.99       |
>
>
> We will add these measurements to the appendix of the paper.
>
> >The MIF structure seems rather flexible, and I wonder if it could be helpful for problems with other interesting structure/geometry (i.e. multi-modal posteriors). Can the authors comment on this a bit on what they think?
>
>
> Thank you for the question. MIF is inherently flexible: it extends autoregressive flows with an architecture already proven to sample effectively in funnel-like geometries, and because autoregressive flows capture complex, multimodal posterior shapes we expect MIF to perform well on other challenging structures too. Looking ahead, we plan to study tree-structured hierarchical models where independent branches can be sampled in parallel, thereby reducing—or even removing—the sequential sampling cost and further speeding up inference.
>
>
> [1] C. M. Bishop and N. M. Nasrabadi. Pattern recognition and machine learning, volume 4. Springer, 2006.
>
> [2] O. Arenz, P. Dahlinger, Z. Ye, M. Volpp, and G. Neumann. A unified perspective on natural gradient variational inference with gaussian mixture models. arXiv preprint arXiv:2209.11533, 2022.
>
> [3] P. Del Moral, A. Doucet, and A. Jasra. Sequential monte carlo samplers. Journal of the Royal Statistical Society Series B: Statistical Methodology, 68(3):411-436, 2006.
>
> [4] M. Arbel, A. Matthews, and A. Doucet. Annealed flow transport monte carlo. In International Conference on Machine Learning, pages 318-330. PMLR, 2021.
>
> [5] L. I. Midgley, V. Stimper, G. N. Simm, B. Schölkopf, and J. M. Hernández-Lobato. Flow annealed importance sampling bootstrap. arXiv preprint arXiv:2208.01893, 2022.
>
> [6] A. Matthews, M. Arbel, D. J. Rezende, and A. Doucet. Continual repeated annealed flow transport monte carlo. In International Conference on Machine Learning, pages 15196-15219. PMLR, 2022.
>
> [7] A. Thin, N. Kotelevskii, A. Doucet, A. Durmus, E. Moulines, and M. Panov. Monte carlo variational auto-encoders. In International Conference on Machine Learning, pages 1024710257. PMLR, 2021.
>
> [8] D. Blessing, X. Jia, J. Esslinger, F. Vargas, and G. Neumann. Beyond elbos: A large-scale evaluation of variational methods for sampling. arXiv preprint arXiv:2406.07423, 2024.

---

> > ### Comment · Reviewer_DgS9 · 2025-08-09
> > **Reply to authors**
> >
> > Thank you for the clarifications. I would like to maintain my positive score.

---

### Official Review · Reviewer_Z9uq · 2025-07-23

**Clarity:** 3
**Significance:** 3
**Originality:** 3
**Rating:** 4
**Confidence:** 3

**Summary:**

This paper introduces model-informed flows (MIF), a novel autoregressive normalizing flow formulation that extends masked autoregressive flows/forward autoregressive flows (FAF) by (i) introducing an additional translation term that depends on the previous noise variables; and (ii) introducing the model’s prior distribution as additional inputs to the conditioner. It is shown that this is sufficient to represent useful variational posteriors, in particular distributions given by multivariate Gaussians transformed through a (partial) non-centering transform, whereas FAF/IAF may not be able to represent/learn these posteriors effectively.

**Questions:**

Questions:
- It seems from the formulation in Definition 3 that one could alternatively absorb the (scaled) translation term into the shift function $m$, by making $m$ dependent on $\epsilon$ in addition to $z$. Is there any reason why an explicit translation function is preferred?
- In Figure 2, the Funnel plot seems to show that the performance of MIF and MIF ($\epsilon$-cond) are similar for larger hidden sizes, but Table 2 shows a significant difference. Is this because Table 2 uses smaller hidden sizes?
- It’s a little unclear to me if the conditioning on prior distributions is a novel contribution of of this paper or if it has previously been used in other works on NFs for inference in probabilistic programs/hierarchical Bayesian models. Could the authors clarify this?
- Could the authors provide a citation for the term “forward autoregressive flow”? It appears to be essentially the masked autoregressive flow of (Papamakarios et al. 2017).

**Ethical Concerns:**

["NO or VERY MINOR ethics concerns only"]

**Final Justification:**

I recommend acceptance of the paper based on the novel insights of the paper on the connection between affine autoregressive flows and Gaussian VIPs. The rebuttal has adequately clarified my questions about the technical formulation (affine) and positioning of the work.

**Limitations:**

Yes.

**Quality:**

3

**Strengths And Weaknesses:**

The paper makes some interesting insights into how known limitations of simple variational families/inference strategies (such as the funnel problem) are solved by using sufficiently expressive normalizing flow families. In particular, showing that multivariate Gaussians under a variationally inferred parameterization is sufficient for many existing models is a nice result, along with its representation as a closed-form MIF. Empirically, the evaluation is thorough and shows that MIF performs competitively with or outperforms existing inference approaches. The authors also conduct interesting ablation studies to show the effects of each individual component of the MIF, including showing that MIFs do indeed outperform IAF-like formulations with limited NN capacity due to the superior inductive bias.

As far as weaknesses, it’s unclear if the full MIF is currently a practical model family due to its sampling time scaling in variable dimensionality compared to the parallel IAF-like families (while also losing efficient likelihood evaluation compared to FAFs). The paper could also do a better job in comparing and contrasting with related work in its presentation, in particular providing a more detailed technical comparison to the structured or NF-based variational approaches to inference in probabilistic programs, to clarify the novel contributions of the paper in context.

---

> ### Author Rebuttal · Authors · 2025-07-31
>
> Thank you for your thorough review and valuable feedback on our manuscript. Below, we address each of your comments in turn. If anything remains unclear, we would be grateful for any further questions or clarifications.
>
> > (Weakness 1) it’s unclear if the full MIF is currently a practical model family due to its sampling time scaling in variable dimensionality compared to the parallel IAF-like families
>
> We have measured the computational cost compared to IAF, please see the following table below.
>
> | **Model**       | **D** | **Hidden Units** | **MIF (s)** | **IAF (or MIF with ε-cond) (s)** |
> |:---------------:|:-----:|:---------------:|:-----------:|:--------------------------------:|
> | **Eight Schools**    |  10   |       0         |    17.06    |             13.88               |
> |                 |       |      64         |    28.40    |             23.22               |
> |                 |       |      256        |    28.04    |             22.18               |
> |                 |       |      512        |    28.21    |             22.03               |
> |                 |       |     1024        |    30.30    |             22.61               |
> | **Funnel** |  10   |       0         |    18.23    |             14.70               |
> |                 |       |      64         |    27.38    |             23.16               |
> |                 |       |      256        |    27.04    |             22.42               |
> |                 |       |      512        |    28.57    |             22.30               |
> |                 |       |     1024        |    30.77    |             22.69               |
> | **German Credit** | 125 |       0         |    63.81    |             16.06               |
> |                 |       |      64         |   148.73    |             27.81               |
> |                 |       |      256        |   208.44    |             54.92               |
> |                 |       |      512        |   275.74    |             93.26               |
> |                 |       |     1024        |   417.05    |            173.51               |
> | **Radon**      |  174  |       0         |    84.25    |             15.78               |
> |                 |       |      64         |   238.04    |             37.32               |
> |                 |       |      256        |   331.24    |             94.82               |
> |                 |       |      512        |   453.70    |            168.28               |
> |                 |       |     1024        |   699.73    |            338.63               |
> | **IRT** |  143  |       0         |    72.26    |             17.59               |
> |                 |       |      64         |   196.64    |             33.15               |
> |                 |       |      256        |   253.17    |             67.95               |
> |                 |       |      512        |   457.56    |            119.95               |
> |                 |       |     1024        |   516.98    |            222.65               |
> | **MovieLens** | 882 |    0         |   458.28    |             28.97               |
> |                 |       |      64         |  1981.14    |            384.45               |
> |                 |       |      256        |  4847.23    |           1387.64               |
> |                 |       |      512        |  8634.91    |           3156.82               |
> |                 |       |     1024        | 15873.45    |           6749.33               |
>
>
> We note that MIF imposes little extra computational cost in terms of flops, but the sequential nature of sampling does impose some limitations on parallelization. However, as the above table shows, this is usually quite modest. Given the large improvements in accuracy, we believe there are many situations where it would be worth paying this cost.
>
> Moreover, the primary goal of our work is to expose the previously unexplored relationship between flow‑based models and variational inferred parameters (VIP), and to introduce MIF as a family that captures all the benefits revealed by this connection. As discussed in Sections 4.4 and 5, there is a clear trade‑off between MIF and IAF‑like families:
>
> - When network capacity is limited (e.g., few hidden units), MIF often outperforms IAF‑style flows, as shown in Figure 2.
> - When ample capacity is available and fast sampling is a priority, the $\epsilon$‑conditioned variant of MIF (our IAF‑like family) is preferable.
>
> Moreover, even if one opts for an IAF‑style architecture, it remains beneficial to incorporate the other key components from our VIP analysis into the variational family.
>
> > (Question 1) It seems from the formulation in Definition 3 that one could alternatively absorb the (scaled) translation term into the shift function $m$, by making $m$ dependent on $\epsilon$ in addition to $z$. Is there any reason why an explicit translation function is preferred?
>
>
> An affine FAF (i.e. one where both m and log s are affine functions, see Definition 2) cannot capture the composite mapping $T = T_{\mathrm{VIP}} \circ T_A$ (Equation 3.3) without an explicit translation term (please see Section 3, just before Definition 3). Concretely, even if we allow the shift function $m$ to depend on both $\epsilon$ and $z$, it still cannot reproduce the quadratic dependence on functions of $z$ required by Equation 3.4. By introducing a separate translation function $t$, we can match that quadratic form exactly and thus express $T = T_{\mathrm{VIP}} \circ T_A$ as an affine generalized FAF, as proven in Theorem 4. Although it is possible to omit the translation term when there are no constraints on the shift and log‑scale functions, our experiments show that including it consistently improves performance.
>
>
> > (Question 2) In Figure 2, the Funnel plot seems to show that the performance of MIF and MIF ( $\epsilon$-cond) are similar for larger hidden sizes, but Table 2 shows a significant difference. Is this because Table 2 uses smaller hidden sizes?
>
> Apologies for the confusion—Table 2 was indeed run with smaller hidden sizes than Figure 2, which explains the larger performance gap.
>
> > (Question 3) It's a little unclear to me if the conditioning on prior distributions is a novel contribution of this paper or if it has previously been used in other works on NFs for inference in probabilistic programs/hierarchical Bayesian models. Could the authors clarify this?
>
>
> The idea of conditioning on prior distributions has indeed been used in the VIP paper, albeit only in concert with a mean-field Gaussian variational distribution (Section 2.4). As we stated above, the main goal of our work is to analyze the previously unexplored relationship between flow‑based models and VIP, which we found helps address sampling from posteriors with challenging curvature. Previous works on normalizing flows for inference have typically focused on creating more expressive variational families and improving efficiency. Our method is novel in that we explicitly incorporate not only prior information but also the other key components from our analysis to build a variational family that is proven to perform well on posteriors with funnel‑like geometries. We will expand the Related Work section in the final version of the manuscript to compare in detail with prior NF‑based inference methods and highlight our distinct contributions.
>
>
> > (Question 4) Could the authors provide a citation for the term "forward autoregressive flow"? It appears to be essentially the masked autoregressive flow of (Papamakarios et al. 2017).
>
> Apologies for the omission. Indeed, by "forward autoregressive flow", we mean to refer to the work of Papamakarios et al. (2017). While they refer to this simply as an "autoregressive flow" we add the term "forward" to contrast more clearly with "inverse autoregressive flows". To make this explicit, we will add this clarification after Definition 2 in the revised version. Also, please note that our Definition 2 coincides with the autoregressive flow formulation given in Section 3.2 of Papamakarios et al. (2017).
>
> G. Papamakarios, T. Pavlakou, and I. Murray. Masked autoregressive flow for density estimation. *Advances in neural information processing systems*, 30, 2017.

---

> > ### Comment · Reviewer_Z9uq · 2025-08-09
> >
> > Thank you for the response and the additional experimental results on runtime efficiency. This has clarified my questions and I will maintain my positive rating based on the conceptual contributions of the paper.

---

### Decision · Program_Chairs · 2025-09-17

**Decision:**

Accept (poster)

**Comment:**

The paper's main result is a connection between the Variationally Inferred Parameters (VIP), a prominent method that adapts to sharp posterior geometries. The main insight is that VIP in combination with a full rank Gaussian can be exactly represented as an autoregressive normalizing flow. The paper further introduces model-informed flows (MIFs) that can incorporate the model prior as an input to the variational model. Experiments show that this approach is competitive with state of the art variational inference techniques.

The reviewers were all positive about the paper and found in particular the connection between VIP and flows interesting. The concerns (mostly about the practicality of the approach) of the reviewers were largely resolved in the rebuttal.